# Combinatorial bZIP dimers display complex DNA-binding specificity landscapes

José A Rodríguez-Martínez[1†], Aaron W Reinke[2†], Devesh Bhimsaria[1,3†], Amy E Keating[2,4], Aseem Z Ansari[1,5*]

[1]Department of Biochemistry, University of Wisconsin-Madison, Madison, United States; [2]Department of Biology, Massachusetts Institute of Technology, Cambridge, United States; [3]Department of Electrical and Computer Engineering, University of Wisconsin-Madison, Madison, Unites States; [4]Department of Biological Engineering, Massachusetts Institute of Technology, Cambridge, United States; [5]The Genome Center of Wisconsin, University of Wisconsin-Madison, Madison, United States

**Abstract** How transcription factor dimerization impacts DNA-binding specificity is poorly understood. Guided by protein dimerization properties, we examined DNA binding specificities of 270 human bZIP pairs. DNA interactomes of 80 heterodimers and 22 homodimers revealed that 72% of heterodimer motifs correspond to conjoined half-sites preferred by partnering monomers. Remarkably, the remaining motifs are composed of variably-spaced half-sites (12%) or 'emergent' sites (16%) that cannot be readily inferred from half-site preferences of partnering monomers. These binding sites were biochemically validated by EMSA-FRET analysis and validated in vivo by ChIP-seq data from human cell lines. Focusing on ATF3, we observed distinct cognate site preferences conferred by different bZIP partners, and demonstrated that genome-wide binding of ATF3 is best explained by considering many dimers in which it participates. Importantly, our compendium of bZIP-DNA interactomes predicted bZIP binding to 156 disease associated SNPs, of which only 20 were previously annotated with known bZIP motifs.

*For correspondence: azansari@ wisc.edu

[†]These authors contributed equally to this work

## Introduction

Multiple sequence-specific transcription factors (TFs) converge at enhancers and promoters to control the expression of genes (*Ptashne and Gann, 2002*). Such TF assemblages permit integration of multiple cellular signals to regulate targeted gene networks (*Ciofani et al., 2012*; *Kittler et al., 2013*; *Xie et al., 2013*). The combinatorial use of a limited number of TFs provides the means to finely control the complex cellular transcriptome. The ability of a given TF to interact with different partners expands the DNA targeting repertoire. It also enhance specificity by focusing combinations of factors to a more specific set of regulatory sites across the genome. Different partners also alter the regulatory potential of a given TF, at times completely altering its regulatory output such that the factor switches from an activator to a repressor of transcription depending on the binding partners with which it associates (*Ptashne and Gann, 2002*).

The bZIP class of human TFs is well suited to play a role in signal integration and combinatorial transcriptional control (*Lamb and McKnight, 1991*; *Miller, 2009*; *Tsukada et al., 2011*). bZIPs bind DNA as either homo- or heterodimers; the discovery in 1988 that JUN and FOS could bind to DNA as a heterodimer immediately suggested the potential for combinatorial regulation by this family (*Franza et al., 1988*; *Lamb and McKnight, 1991*). Interestingly, the human bZIP network displays greater ability to form heterodimers compared to simpler eukaryotes, suggesting that more complex

**eLife digest** Most cells in our body contain the same DNA blueprint, which encodes all the genes needed for every process in the body. However, only certain genes need to be active in a particular cell at any given time. Proteins known as transcription factors control the activity of genes by binding to DNA near the start of the genes and switching genes on or off as required. Often transcription factors work together to regulate specific genes in response to signals from other cells or the environment. Failure to control the activity of genes can give rise to cancer, diabetes and a wide array of other diseases.

The bZIP family of transcription factors regulates the activities of many genes. These transcription factors work in pairs to bind a specific DNA site. They either partner with an identical molecule or with a different bZIP transcription factor. Different combinations of bZIP pairs prefer to bind different stretches of DNA. Except for a few examples, it is not yet understood how bZIP pairs work together to find the right target DNA.

Rodriguez-Martinez, Reinke, Bhimsaria et al. have identified all of the DNA sites that 102 pairs of human bZIP transcription factors can bind to. The experiments show that over two thirds of the bZIP pairs bind to DNA sequences that each individual partner prefers. However, in many cases, the choice of a partner can change the DNA sequence that the pair targets in a manner that could not have been predicted based on the preferences of each partner alone. This suggests that, by pairing up, bZIP transcription factors are able to change their preferences for which location they target in the DNA. The experiments also show that many of the 102 pairs could bind to more than one type of site. Thus, the ability of bZIP proteins to interact with different partners greatly expands the locations on genomic DNA from which they can regulate the activity of different genes.

DNA sequences vary between different individuals and some variants can predispose individuals to certain diseases. Rodriguez-Martinez et al. found that bZIP pairs can bind to over a hundred DNA variants that are associated with disease. The next challenge is to find out how specific variations in DNA can lead to the formation or elimination of bZIP binding sites that cause disease. In the future, DNA editing methods may make it possible to specifically fix such changes in our genomes to reduce the risk of disease.

combinatorial regulation may contribute to organismal complexity (*Reinke et al., 2013*). bZIP proteins also interact with other classes of TFs to stabilize higher order oligomers at enhancers (*Jain et al., 1992*; *Murphy et al., 2013*; *Thanos and Maniatis, 1995*). Their role in nucleating such multi-factor complexes is supported by the observation that certain bZIP dimers such as AP-1 (FOS•JUN) and CEBPA can function as 'pioneer' factors that bind inaccessible chromatin and enable the assembly of other TFs at regulatory sites (*Biddie et al., 2011*; *Collins et al., 2014*). Notably, the choice of dimerizing partner not only impacts DNA recognition properties but can also influence regulatory function of a given bZIP. For example, ATF3 homodimer acts as a repressor, whereas ATF3•JUN activates transcription (*Hsu et al., 1992*). As a class, bZIPs regulate diverse biological phenomena ranging from response to stress at the cellular level, organ development at the tissue level and viral defense, circadian patterning, memory formation, and ageing at an organismal level (*Costa et al., 2003*; *Herdegen and Waetzig, 2001*; *Jung and Kwak, 2010*; *Male et al., 2012*). Given their central role in various processes, mutations in bZIP proteins are implicated in the etiology of diseases ranging from cancer and diabetes to neuronal malfunction, developmental defects and behavioral dysfunction (*Lopez-Bergami et al., 2010*; *Tsukada et al., 2011*).

Fifty-three bZIPs encoded by the human genome can be grouped into 21 families, and as homodimers they are known to bind at least six distinct classes of DNA motifs, including sites labeled as TRE, CRE, CRE-L, CAAT, PAR, and MARE (*Figure 1*) (*Deppmann et al., 2006*; *Jolma et al., 2013*). In 1991, *Hai and Curran (1991)* showed that some heterodimers have DNA-binding specificities that are distinct from those of each partnering bZIP. For example, JUN•ATF2 heterodimer binds to a cognate site in the ENK2 promoter that is not bound by either JUN•JUN or ATF2•ATF2 homodimers. However, the past 20 years have provided only a handful of additional examples of how bZIP heterodimerization influences DNA-binding specificity (*Cohen et al., 2015*; *Jolma et al., 2015*;

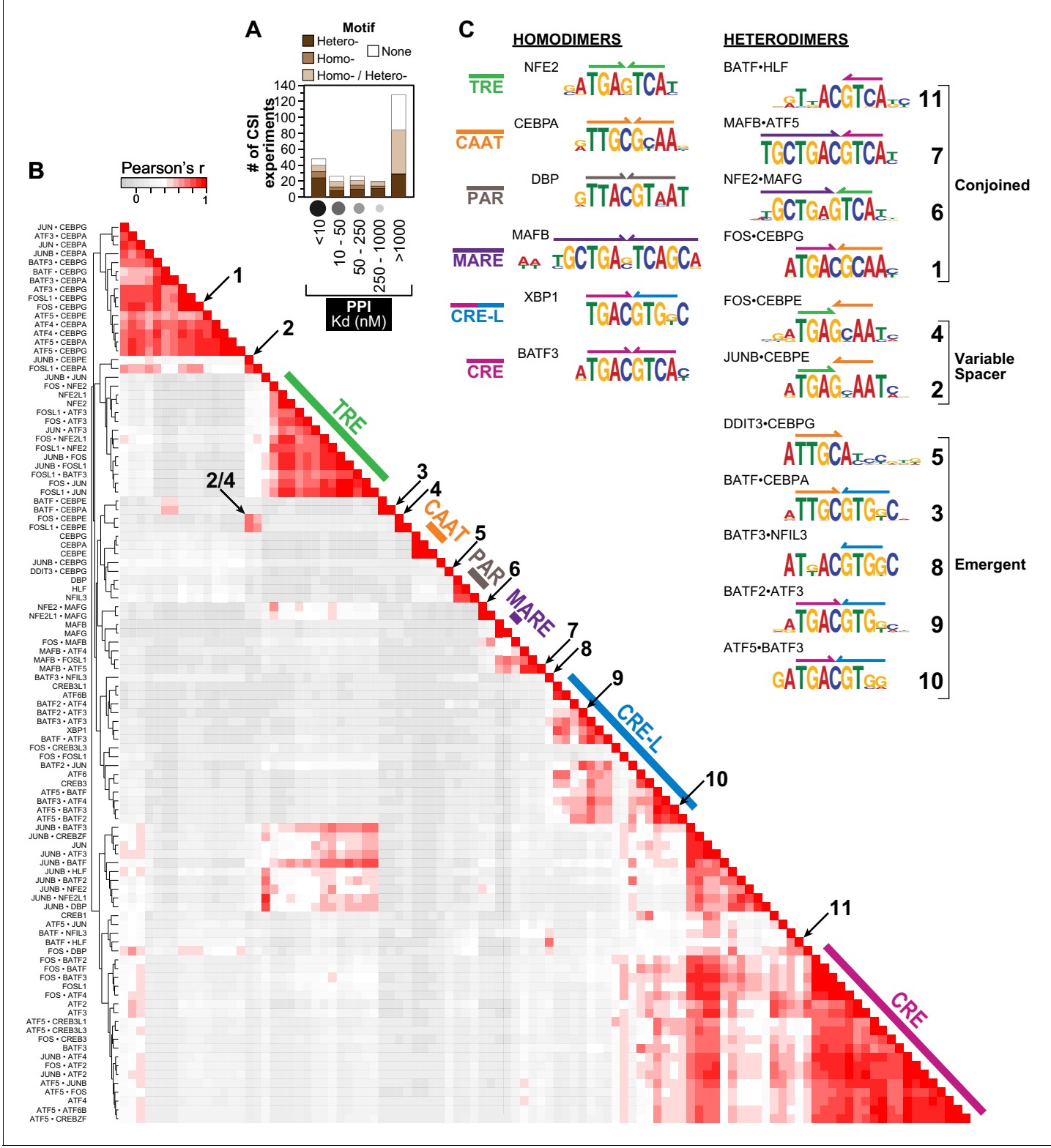

**Figure 1.** Overview of human bZIP homodimer and heterodimer DNA-binding specificities. (**A**) Summary of SELEX-seq results categorized by protein-protein interaction (PPI) affinity (*Reinke et al., 2013*). Specificity profiles were classified as resulting in a motif arising from DNA binding by either a homodimer (brown) or a heterodimer (dark brown), or not resulting in a motif (white). Some profiles could not be unambiguously assigned to a homo vs. heterodimer (light brown). (**B**) Pairwise comparisons of the DNA-binding preferences of 102 bZIP dimers (22 homodimers and 80 heterodimers) using the z-scores for 1222 unique 10 bp sequences corresponding to the 50 top ranked sequences for each dimer. Throughout the paper, the biotinylated

*Figure 1 continued on next page*

*Figure 1 continued*

bZIP is listed first when describing a heterodimer. (**C**) Representative motifs bound by bZIP homodimers and heterodimers reported in this study. Heterodimer motifs were grouped as Conjoined, Variable spacer, and Emergent. The color code defined here for half sites (colored arrows above motifs) is used throughout the figures.

The following source data and figure supplements are available for figure 1:

**Source data 1.** Data for *Figure 1C*.

**Figure supplement 1.** Cognate site identification by SELEX-sequencing.

**Figure supplement 2.** ATF3 CSI Intensity (z-score) correlates with equilibrium association constant.

**Figure supplement 3.** Pairwise comparison of bZIP homodimers reported in this study and bZIP dimers reported by Jolma et al.

**Figure supplement 4.** bZIP heterodimer specificity.

**Figure supplement 5.** DNA sequence preferences for FOS•CEBPE, FOS•CEBPG, FOSL1•CEBPE, and FOSL1•CEBPG.

*Vinson et al., 1993*; *Yamamoto et al., 2006*). Central questions about bZIP transcription factors remain unanswered: What is the influence of protein dimerization on DNA binding? Does DNA stabilize dimer formation? Which protein dimers can bind DNA? Which sequences do they bind? And, how do bZIP-binding sites contribute to cellular function and the etiology of various diseases?

Resolving these issues requires systematic examination of the DNA-binding specificities of bZIP homo- and heterodimers. Fifty-three human bZIP proteins can potentially form as many as 1431 distinct dimers. Quantitative experiments using fluorescence resonance energy transfer (FRET) in solution indicated that ~ 30% of all possible bZIP dimers form in the absence of DNA. Most bZIPs can form dimers with different partners, potentially greatly expanding the repertoire of cognate sites that might be targeted by different heterodimers (*Reinke et al., 2013*). We used this protein-protein interaction (PPI) dataset to prioritize 270 bZIP dimers for bZIP-DNA interactome studies and to apply FRET-based methods to distinguish DNA-bound heterodimers from homodimers.

Insights that emerged from our compendium of bZIP-DNA interactomes include: (i) identification of new bZIP cognate sites, (ii) evidence for three classes of heterodimer-binding sites (conjoined half-sites, variably-spaced half-sites, and unpredicted emergent cognate sites), (iii) ability of individual bZIP heterodimers to target a range of binding sites, (iv) evidence for varying heterodimer selectivity between distinct sequences currently classified as a single consensus motif, (v) improved ability to account for in vivo genome-wide occupancy of heterodimers, and (vi) identification of bZIP cognate sites at 156 SNPs linked to human diseases and quantitative traits. DNA sequence preferences of bZIP heterodimers reported here serve as a valuable resource for many purposes including, but not limited to, evaluating potential bZIP dimer binding at genomic binding sites, providing hypotheses about mechanisms underlying the etiology of disease-linked SNPs, and predicting binding specificities of heterodimeric bZIPs from other species.

## Results

### Comprehensive bZIP-DNA interactomes

To elucidate DNA sequence-recognition properties of TFs that form obligate dimers, we examined 270 pairs of purified human bZIP proteins. These pairs were composed from 36 bZIP proteins representing 21 bZIP families and encompassing the diversity of all 53 bZIPs encoded in the human genome (*Supplementary file 1A*). Given the 666 potential dimeric pairs that can be formed with 36 bZIPs, we used biophysically measured PPIs to prioritize the dimers that were examined (*Reinke et al., 2013*). We selected 126 pairs (97 hetero- and 29 homodimer) that form stable dimers with PPI dissociation constants ($K_d$) less than 1 μM at 21°C in the absence of DNA. In addition, we tested 144 (137 hetero- and seven homo-) dimer combinations that do not stably associate in

solution in the absence of DNA ($K_d$ >1 µM at 21°C). For most TFs, including the bZIP class, the DNA specificity of the isolated DNA binding domain (DBD) is typically indistinguishable from the full-length factor (*Jolma et al., 2013*). Therefore, we focused our efforts on the bZIP domain, which comprises the basic region that binds DNA and the leucine zipper dimerization module that forms a coiled-coil. Recombinant proteins, overexpressed in bacteria, were purified to homogeneity. Two versions of each protein were made, one conjugated to biotin at the carboxyl terminal and the other without. This set of highly purified DNA binding proteins enabled examination of the innate DNA-binding sequence specificity of 36 representative human bZIPs. Individual bZIP partners were mixed in 3:1 molar ratios with the biotinylated partner at the lower concentration; affinity purification of the protein-DNA complexes using the less abundant biotinylated partner enriched for heterodimers. To additionally favor examination of the heterodimer, whenever possible, the interaction partner with the weaker homodimer was biotinylated and used for isolating protein-DNA complexes. Each protein dimer is denoted by a dot between each monomer – for example, JUN•ATF3. The DNA binding sites are indicated either by a specific sequence or their classical designations, such as CRE or CAAT, or by half-sites connected by a hyphen, such as CRE-CAAT.

DNA-binding specificity of the pairs was queried using systematic evolution of ligands by exponential enrichment coupled to deep sequencing (SELEX-seq) (*Figure 1—figure supplement 1*) (*Jolma et al., 2010*; *Slattery et al., 2011*; *Tietjen et al., 2011*; *Zhao et al., 2009*; *Zykovich et al., 2009*). In our cognate site identification (CSI) effort using SELEX-seq, a DNA library spanning the entire sequence space of a 20-mer ($10^{12}$ different sequence permutations) was independently incubated with each of the 270 different bZIP pairs (234 hetero- and 36 homodimers), and protein-bound DNA sequences were enriched, amplified, and re-probed for an additional two cycles to further enrich cognate sites over non-cognate sites that comprise the majority of the starting library. The starting DNA library as well as selectively enriched sequences were barcoded and sequenced using massively parallel DNA sequencing methods. The CSI intensity, corresponding to the z-score for the enrichment of a sequence, was computed for each 10-mer as described in the methods. Repeated experiments demonstrated an average Pearson's correlation r = 0.8 ± 0.1 between CSI intensities from replicates (*Figure 1—figure supplement 1*). The CSI intensity (z-score) correlates with the binding affinity for a particular sequence (*Figure 1—figure supplement 2*) (*Carlson et al., 2010*; *Puckett et al., 2007*; *Tietjen et al., 2011*). In three cases, reciprocal biotinylation of each partner was performed to ensure that the choice of partner did not skew the results (Pearson's correlation for comparing experiments was r = 0.89–0.98; *Figure 1—figure supplement 1C*).

Among the 270 pairs tested were 12 homodimers that had been previously examined by other groups (*Jolma et al., 2013*). Overall, we found excellent agreement between the cognate sites identified using highly purified bZIP modules in our study versus full-length proteins in unpurified cell lysates in other studies, with only a few inconsistent examples that can be seen in *Figure 1—figure supplement 3A* (e.g. MAFB, NFE2, ATF4). Interestingly, we found that the previously reported DNA specificity of ATF4 has a higher correlation with the specificity of ATF4•CEBPG heterodimer identified in this report than with the specificity of the ATF4 homodimer (*Figure 1—figure supplement 3B*), suggesting that CEBPG possibly formed a complex with ATF4 in the cell lysates used in the prior study (*Jolma et al., 2013*). This observation highlights the advantage of using highly purified proteins over cell lysates and validates our focus on the bZIP domain to capture the innate specificity of this class of transcription factors that bind DNA as obligate dimers.

Overall, 30 out of the 36 bZIP proteins tested in this study enriched specific DNA sequences as part of at least one dimer. bZIPs that are not known to bind DNA as homodimers did not yield cognate sites in our studies (e.g. JUNB and FOS) (*Deng and Karin, 1993*; *Hai and Curran, 1991*). 73 of 126 (58%) bZIP pairs that dimerize in the absence of DNA yielded specific cognate sites (*Figure 1A*). Surprisingly, 29 of the 144 (20%) bZIP pairs that do not stably associate in the absence of DNA ($K_d$ >1 µM at 21°C) yielded evidence of sequence-specific binding to DNA, indicating that PPIs were stabilized by binding to specific DNA sites (*Figure 1A*; *Supplementary file 1C*). This finding has important implications, given that the majority of the potential bZIP PPI space consists of protein pairs that do not associate strongly in the absence of DNA (*Reinke et al., 2013*).

## Conjoined, variably-spaced, and emergent cognate sites bound by heterodimers

For 184 bZIP-DNA interactomes that showed evidence for an enriched motif, we computationally parsed and retained datasets that could be attributed with high confidence to 80 heterodimers and 22 homodimers (Materials and methods). We assigned a specificity profile to a heterodimer when it bound sequences that were significantly different (t-test p<0.05) from the sequences preferred by the homodimer of the biotinylated bZIP (e.g. ATF4 vs. ATF4•CEBPA, r = 0.1; *Figure 1—figure supplement 4*) or when the biotinylated bZIP did not bind specific DNA sequences as a homodimer (e.g. FOS and JUNB).

Of the 22 homodimers, comprehensive DNA-binding specificity is reported for the first time for human ATF2, ATF3, ATF6, ATF6B, CEBPA, CREB1, FOSL1, JUN, MAFB, and NFE2L1. Hierarchical clustering of the 102 bZIP-DNA interactomes readily identified six previously known classes of bZIP-binding sites (TRE, CAAT, PAR, MARE, CRE, and CRE-L) (*Figure 1B–C*). Notably, several bZIP homodimers (ATF6, ATF6B, CREB3L1, and JUN) enriched more than one motif (*Supplementary file 2*). Examining the cognate sites bound by heterodimers highlighted the ability of some heterodimers to bind homodimer motifs as well as a range of other heterodimer-specific motifs. Such binding to multiple motifs is reminiscent of previous studies that reported bZIP dimers binding to different sites with different affinities (*Badis et al., 2009*; *Kim and Struhl, 1995*; *König and Richmond, 1993*). Interestingly, several heterodimers that bind classic bZIP homodimer motifs such as TRE, CRE-L, or CRE displayed clear differences in their preference for a subset of sequences categorized under a single consensus motif (e.g. compare motifs **9** and **10** in *Figure 1* with the CRE-L site). This was also true for different homodimers (e.g. compare CRE-L binding profiles for ATF6, CREB3, CREB3L1, and XBP1 in *Supplementary file 2*). Thus, the binding data reported here reveal a sequence sub-structure to classic consensus motifs. Moreover, the sub-structure highlights differences in DNA-binding specificity between closely related dimers.

Three classes of bZIP heterodimer motifs were identified and are illustrated in *Figure 1C*: '*Conjoined*' sites for which half-sites preferred by each contributing monomer are juxtaposed (such as the CRE-CAAT site represented by motif **1**, or the MARE-CRE site of motif **7**), '*Variably-spaced*' sites for which half-sites overlap (as is the case in motifs **2** and **4**), and '*Emergent*' sites for which binding preferences could not have been readily inferred based on the half-site preferences of each partner (motifs **3**, **5**, **8**, **9**, and **10**). In other words, an *emergent* site arises as a consequence of heterodimer formation and is not simply comprised of the conjoined or variably-spaced half-sites preferred by each monomer. An elegant study of Hox-Exd heterodimers identified 'latent' sites that were preferred by different Hox factors when they bound DNA in conjunction with Exd (*Slattery et al., 2011*). Preferences for different sequences at the interface of half-sites or sequences flanking the half sites were observed for different classes of Hox-Exd heterodimers. In our studies, we observed a change in half-site preference of certain bZIPs when they bound DNA as heterodimers. In some instances, homodimers bound with low affinity to sites that emerged as high-affinity sites in the context of a heterodimer, whereas in other cases entirely new site preferences emerged. We classified such newly acquired binding preferences as *emergent* sites because they are not readily inferred from the binding preferences of homodimers.

While a large fraction of heterodimers bind conjoined sites, it was surprising to find that closely related heterodimers such as FOS•CEBPG and FOS•CEBPE preferred different arrangements of half-sites, with the former heterodimer preferring the 8 bp conjoined CRE-CAAT site (motif **1** $^{5'}$TGACG-CAA$^{3'}$) and the latter preferring the 7 bp variably-spaced TRE-CAAT site (motif **4** $^{5'}$TGAGCAA$^{3'}$). *Figure 1—figure supplement 5* highlights the unexpectedly poor correlation between the binding preferences of these two heterodimers and between the binding preferences of FOSL1•CEBPG and FOSL1•CEBPE. Similarly, other heterodimers bound both conjoined and variably spaced motifs (see JUNB•ATF3 and MAFB•ATF5 in *Supplementary file 2*); however, the preference for one arrangement over the other was not amenable to predictions based on the binding preferences of each contributing partner of the heterodimer.

Emergent sites pose a particular challenge for current models of DNA binding site predictions that are based on protein homology (*Weirauch et al., 2014*). Emergent cognate sites for heterodimers can be subdivided into two categories: (i) 'gain-of-specificity' motifs that display a change in half-site preferences for a bZIP or (ii) motifs that display a 'loss-of-specificity' for one half-site. An

example of the first category includes a switch in the half-site preferences of BATF family members, from a CRE half-site ($5'$TGAC$^{3'}$) that is preferred in homodimers to a CRE-L ($5'$CCAC$^{3'}$) half-site that is preferred by many BATF-containing heterodimers (compare motifs **3** and **8–10,** and see examples in *Supplementary file 2* such as BATF2•ATF3, BATF2•JUN, BATF3•ATF3, BATF3•ATF4). An example of the second category is DDIT3•CEBPG binding to $5'$ATTGCA$^{3'}$ (motif **5**) (*Ubeda et al., 1996*), with heterodimers displaying no apparent requirement for one half-site. Overall, for the 80 bZIP hetero-dimers with binding motifs reported here, 72% of the motifs can be classified as conjoined, 16% as emergent, and 12% as variably-spaced. Nine out of the 80 heterodimers (11%) enriched two motifs (*Supplementary file 2*). For example, BATF•CEBPG enriched both CRE-CAAT and CRE-L-CAAT motifs.

## Specificity and energy landscapes reveal the entire spectrum of cognate sites bound by heterodimers

To examine the full specificity spectrum of individual bZIP dimers, we displayed DNA binding data as specificity and energy landscapes (SELs) (*Carlson et al., 2010*; *Tietjen et al., 2011*). In a SEL, all possible sequences of a given length are arranged within concentric circles based on their homology to a seed motif. The seed motif is often derived from position weight matrices (PWMs) of the most enriched sequences (*Figure 2A*). The innermost circle contains all sequences that have an exact sequence match to the seed motif (0-mismatch). As each enriched sequence placed in this ring is an exact match to the seed motif, the source of varying CSI intensity (z-score) is the contribution of the sequences flanking the seed motif. The 1-mismatch ring contains all sequences that differ from the seed motif at any one position, or a Hamming distance of one. The subsequent rings, going out-wards, display sequences with increasing number of mismatches to the seed motif. The height and color of each point represents the CSI intensity for the corresponding sequence. As noted above, CSI intensity correlates with binding affinity where measured (*Figure 1—figure supplement 2*) (*Carlson et al., 2010*; *Hauschild et al., 2009*; *Puckett et al., 2007*; *Tietjen et al., 2011*; *Warren et al., 2006*). Although there are far more low-affinity sequences than enriched sequences (as depicted by the illustrative histogram in *Figure 2A*), the moderate-to-low affinity sites (low CSI intensity) are often overlooked by motif searching algorithms. Such sequences readily emerge in SEL display of the entire binding data (*Carlson et al., 2010*; *Tietjen et al., 2011*). In *Figure 2A*, we illus-trate how SELs are built and we note that a SEL can be constructed using any sequence as a seed motif. The choice of a different seed motif simply alters the placement of the sequences on the land-scape without changing the underlying binding preferences of a protein for a given sequence.

SEL plots of 102 bZIP homo- and heterodimers reveal that the impact of flanking sequence con-text and the range of different cognate sites bound by most bZIPs is far richer than might be inferred from motifs represented as PWMs (*Supplementary file 2*). In *Figure 2B*, the SELs of JUN•ATF3 and ATF4•CEBPG illustrate the broader insights that emerged from examining specificity profiles of these two heterodimers. JUN•ATF3 binds a CRE site composed of conjoined half-sites for JUN and ATF3. Visualizing the entire JUN•ATF3-DNA interactome via a SEL shows that the binding of JUN•ATF3 heterodimer to CRE is significantly influenced by the sequence context that flanks the motif (see affinity variations in the 0-mismatch ring). Additionally, the 3-mismatch ring of the SEL identifies several high-intensity peaks corresponding to additional cognate sites. As indicated, one of these is a variably spaced site, and another is an emergent site $5'$TGACGCAT$^{3'}$. Thus, the SEL highlights that this single heterodimer binds multiple classes of cognate sites. On the other hand, the SEL for ATF4•CEBPG shows that the seed motif $5'$ATGCGCAAT$^{3'}$ bound by this heterodimer is relatively insensitive to context effects (0-mismatch ring). The 1-mismatch ring indicates that both half-sites are not equally tolerant of mismatches, with mismatches in the $5'$TGA$^{3'}$ core of the CRE site dramatically reducing binding, whereas the $5'$CAA$^{3'}$ site is tolerant of deviations at the first position of the half-site but sensitive to deviations in the $5'$AA$^{3'}$ positions. Similar insights can be obtained from the SELs for each of the102 bZIP dimers that are reported in *Supplementary file 2*.

Our compendium of SEL plots greatly extend previous reports that bZIP dimers bind a range of sequences with different degrees of affinity (*Badis et al., 2009*; *Kim and Struhl, 1995*; *König and Richmond, 1993*). To examine whether the set of sequences that are pointed out in SELs of JUN•ATF3 and ATF4•CEBPG from *Figure 2B* are bound by homodimers or any bZIP in our compen-dium, we displayed the relative preferences of each dimer for this set of binding sites in a heatmap (*Figure 2C*). Each column displays the relative preference of each of the 102 bZIP dimers for

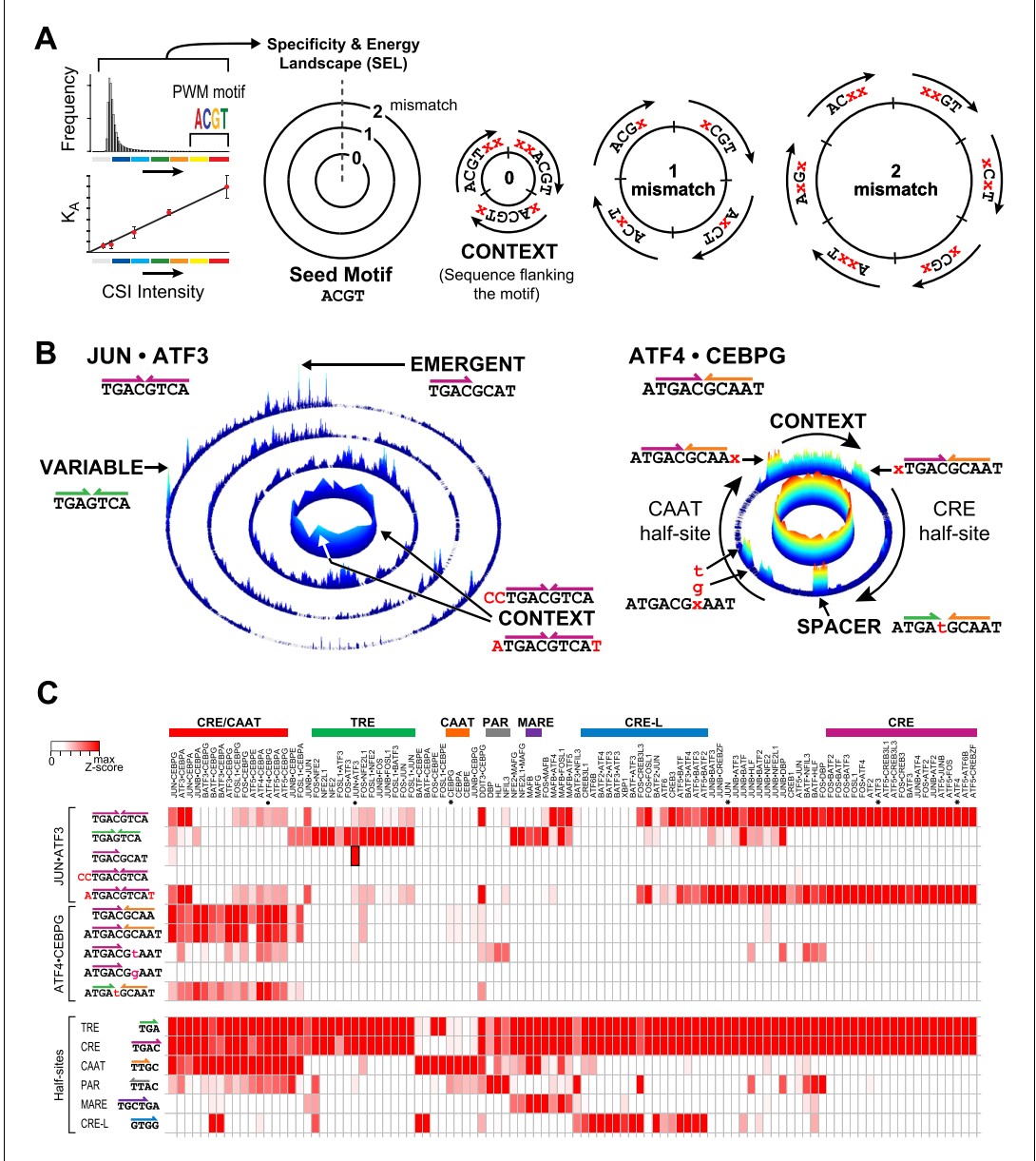

**Figure 2.** Specificity and energy landscapes (SELs) and motifs for bZIP heterodimers. (**A**) SEL displays CSI intensities for all sequence permutations of a given binding site size (k-mers). Sequences are organized with respect to any selected seed motif; however, a k-mer representing PWM-derived motif is typically used. CSI intensities correlate with equilibrium binding affinities. As an example, the arrangement of 6-mer sequences for a simplified 4-mer seed motif is shown. The innermost circle displays the intensities for all sequences that have an exact match to the seed motif (0-mismatch ring). In this ring, sequences are arranged in a clockwise manner with sequences that include residues 5′ of the seed motif at the start, sequences with residues that flank both 5′ and 3′ ends in the middle, and 3′ flanking sequences at the end (context). The subsequent 1-mismatch ring contains the sequences that differ at one position from the seed. The sequences are organized clockwise starting with mismatches at the first position and ending with mismatches at the last position of the motif. Within each sector, the mismatches at a given position (indicated by x) are organized in alphabetical order (A, C, G, and T). The 2-mismatch ring contains all permutations with two positional differences with the seed, similarly ordered. (**B**) Left, SEL for JUN•ATF3 heterodimer using CRE (⁵′TGACGTCA³′) as the seed motif. By displaying the 10 bp sequence space, preferred sequences become apparent. Peaks corresponding to emergent and variably-spaced sites are identified by arrows. Right, SEL displaying 12 bp sequences for ATF4•CEBPG heterodimer using CRE-CAAT (⁵′ATGACGCAAT³′) as the seed motif. (**C**) Heatmap of the relative CSI intensities of 102 bZIP dimers (columns) for the 10 sites highlighted in *Figure 2B* as well as constituent half-sites of the six classic bZIP motifs (rows). Displayed is the maximum CSI intensity of all the 10 mers matching the site. bZIP dimers are listed in the same order as in *Figure 1B*. ATF3, ATF4, CEBPG, and JUN homodimers are marked by asterisks. While bZIPs do not bind as monomers to half-sites, the occurrence of bZIP half-sites within motifs is displayed in the second set of rows to enable comparison between the half-site preferences versus the CSI intensity for motifs that display these half-sites in different combinations or in different contexts.

*Figure 2 continued on next page*

*Figure 2 continued*

The following source data is available for figure 2:

**Source data 1.** Data for *Figure 2C*.

different sequences, including half-sites of all six classical homodimer motifs. An examination of row 3, which displays preferences of all 102 dimers for the emergent site $^{5'}$TGACGCAT$^{3'}$, indicates that this site is highly preferred by JUN•ATF3 and to some extent by JUN•CEBPG but not by homo-dimers formed by ATF3, CEBPG, or JUN (denoted by asterisks). While not as exclusive as the JUN•ATF3 emergent site, the conjoined CRE-CAAT site is primarily targeted by heterodimers formed by the CEBP family of bZIPs. The heatmap does indicate that this heterodimer-preferred site permits low-affinity binding by the CEBPG homodimer. Interestingly, data in row 8 reveals that substituting $^{5'}$CAAT$^{3'}$ with $^{5'}$TAAT$^{3'}$ in the CRE-CAAT site perturbs binding by CEBP heterodimers in a non-uniform manner, unmasking hidden differential sequence preferences of related heterodimers that are opaque to current models that use protein homology to predict cognate site preferences. The C-to-T substitution also expands the repertoire of bZIPs that bind this mutated site. DBP, HLF, and NFIL3 as homo- and hetero- dimers display an unmistakable affinity for this modified CRE-CAAT site that recreates the PAR half-site that is a target of this set of bZIPs. On the other hand, a different substitution at the same position ($^{5'}$GAAT$^{3'}$) dramatically reduces the binding of all bZIPs to this ver-sion of the CRE-CAAT site (row 9). Furthermore, the importance of the sequences flanking a binding site (rows 4 and 5) or the contribution of each half-site to the binding of bZIPs (rows 8–10) is also made evident by the heatmap. In essence, SEL plots alongside comparative heatmaps of affinities of proteins for a range of cognate sites bring a new appreciation for diversity of DNA sequences that can be targeted by a given factor.

## EMSA-FRET analysis to validate heterodimer binding to different cognate sites

To validate the ability of heterodimers to bind cognate sites identified by CSI analysis, we used an electrophoretic mobility shift assay (EMSA) in which a FRET signal distinguished homo- vs. hetero-dimers in protein-DNA complexes (*Figure 3A*; Materials and methods) (*Reinke et al., 2013*). We first used EMSA-FRET to assay bZIP dimers formed by mixing fluorescein and TAMRA labeled versions of 16 proteins drawn from different bZIP families. For 15 homodimers for which we could detect bind-ing to DNA, the mixed-dye homodimer could be easily distinguished from both of the single-dye homodimers, as shown for CEBPG and ATF3 homodimers binding to CAAT and CRE sites, respec-tively (*Figure 3A*). This assay was then used to demonstrate that the ATF3•CEBPG heterodimer bound the conjoined CRE-CAAT site better than either parental homodimer (*Figure 3A*). Further-more, swapping fluorophores did not alter the binding properties of the resulting heterodimer (last panel of *Figure 3A*). DNA fluorescence coincides with the protein FRET signals, confirming that pro-tein-DNA complexes were being observed in the EMSA gels (*Figure 3—figure supplement 1A*).

We used this EMSA-FRET assay to quantify the DNA binding of 83 bZIP homodimers and hetero-dimers comprised of 16 proteins. Each heterodimer was systematically examined with DNA sites that were constructed by conjoining the preferred half-site(s) for each bZIP. *Figure 3B* shows EMSA-FRET data for six heterodimers and corresponding homodimers binding to heterodimer-specific sites (three conjoined sites and three emergent sites). For these sites, the CSI intensity for the heter-odimers is higher than the scores for either of the two contributing homodimers. The EMSA-FRET data demonstrate clearly that neither the JUN nor the ATF3 homodimers associate with the emer-gent site identified for JUN•ATF3 (*Figure 3B* and *Figure 3—figure supplement 2*). Similarly, emer-gent sites identified for ATF4•CEBPA and ATF4•JUN, and several conjoined sites such as TRE-CAAT for ATF3•CEBPA, CRE-L-CAAT for BATF3•CEBPA, and CRE-CRE-L for BATF3•JUN, were validated by EMSA-FRET as *bona fide* heterodimer-specific cognate sites that show weaker binding, or no binding, by the contributing homodimers (*Figure 3B*). EMSA-FRET data also validate the ability of BATF3 to bind emergent CRE-L half-sites as a heterodimer (in addition to the CRE site preferred by the homodimer). The complete EMSA-FRET data are presented in a more compact format in *Fig-ure 3—figure supplement 1*.

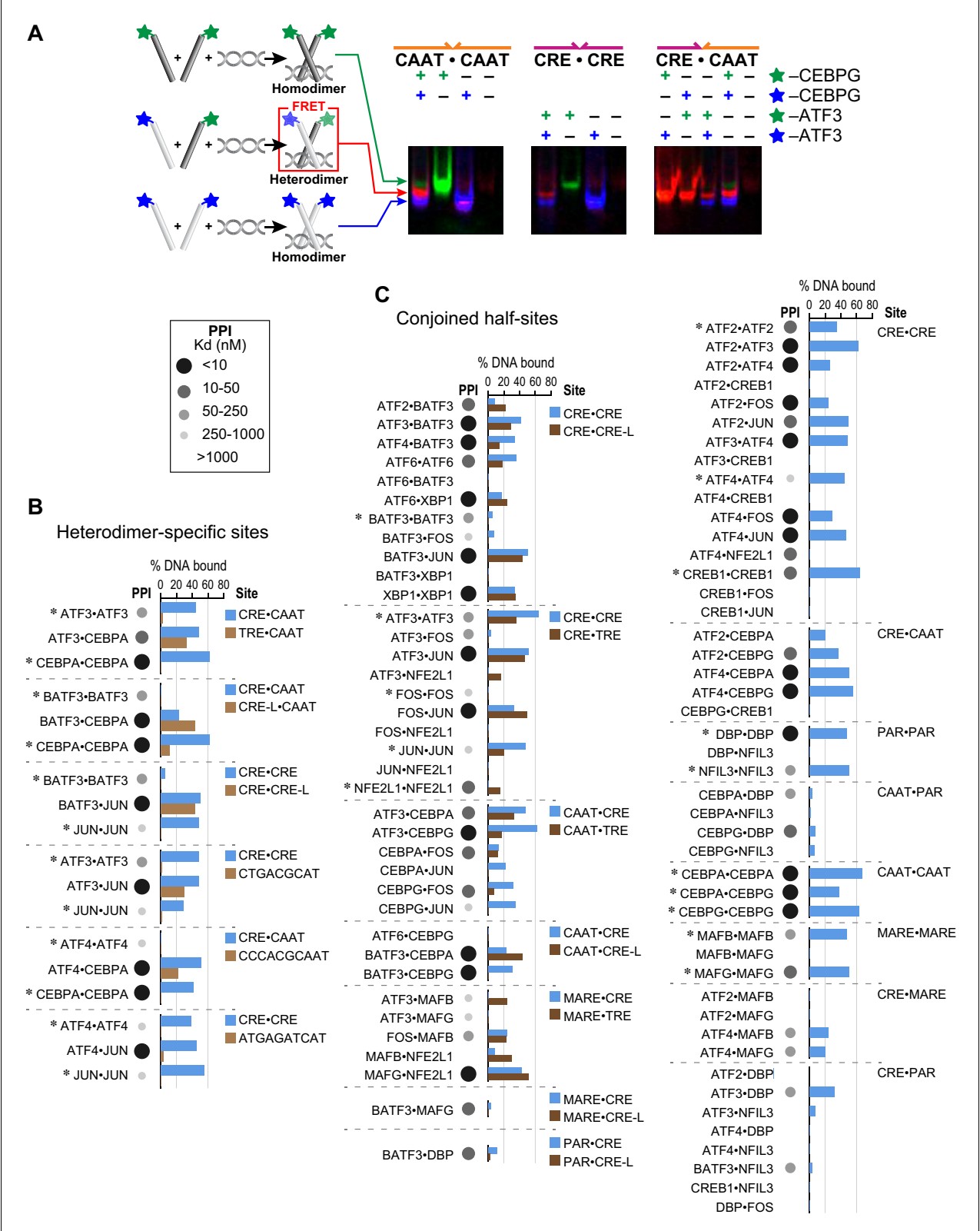

**Figure 3.** Influence of bZIP protein dimerization on DNA binding. (A) EMSA-FRET assay used to quantify bZIP heterodimers and homodimers binding to DNA. Fluorescein and TAMRA are depicted as blue and green stars, respectively. In the EMSA gel, homodimers give rise to pseudo-colored blue (fluorescein) or green (TAMRA) signals, whereas heterodimers give a FRET signal that is pseudo-colored red. (B) EMSA-FRET results for bZIP dimers binding to selected heterodimer-specific emergent sites (brown) and conjoined half-sites (blue). Bar graphs show the percent of the indicated DNA

*Figure 3 continued on next page*

*Figure 3 continued*

oligomer bound by each dimer. The PPI strength of each dimer is indicated with gray-scale circles sized according to the $K_d$ for a given protein-protein interaction. Homodimers are marked with an asterisk (*). (C) EMSA-FRET results for bZIP dimers tested for binding to DNA sites composed of conjoined half-sites. Left, dimers tested against two different sites composed of conjoined half-sites. Right, dimers tested against a single site. Data are displayed as in B.

The following figure supplements are available for figure 3:

**Figure supplement 1.** Influence of bZIP protein dimerization on DNA binding.
**Figure supplement 2.** Heterospecific binding of DNA.

A striking result of our CSI analysis is that conjoined half-sites form a substantive fraction (~70%) of the cognate sites bound by heterodimers. To determine how frequently DNA half-sites derived from homodimer-binding data, when presented as conjoined sites, would bind the corresponding heterodimers, we tested DNA binding by EMSA-FRET for stably interacting bZIP heterodimers (PPI: $K_d$ <1 μM at 21°C), and the corresponding homodimers (*Figure 3C*). Consistent with CSI analysis, 52 out of 56 bZIP pairs that form stable heterodimers bound the DNA site made by conjoining the half-site preferred by each monomer. Specific binding to conjoined sites was also detected for 6 out of 27 (22%) pairs that do not stably associate in the absence of DNA (PPI: $K_d$ >1 μM at 21°C). This fraction is similar to the 20% of bZIP pairs (29 out of 144) that showed sequence-specific DNA binding in SELEX-seq experiments despite their apparent inability to dimerize in the absence of DNA.

## ATF3: a case study of the influence of interacting partners on heterodimer cognate site preferences

Activating Transcription Factor 3 (ATF3) is a member of the CREB/ATF family. Initially identified as a suppressor of inflammation and the adaptive immune response in resting cells, ATF3 is now associated with numerous diseases including a variety of aggressive and widely occurring cancers (*Tanaka et al., 2011*; *Thompson et al., 2009*; *Yin et al., 2008*). ATF3 is able to interact with a large variety of TFs to function as a regulatory hub of cellular adaptive response (*Gilchrist et al., 2006*; *Hai et al., 1999*, *2010*). As a homodimer, ATF binds to CRE sites and represses a wide array of genes (*Hai et al., 1999*, *2010*). However, as a heterodimer with JUN or JUND, ATF3 activates transcription of targeted genes (*Chu et al., 1994*; *Filén et al., 2010*; *Hsu et al., 1992*).

To test the hypothesis that heterodimerization with other bZIPs might alter DNA-binding specificity and possibly genomic targets, we analyzed SELEX-seq for 20 different ATF3 heterodimers spanning the full range of PPI affinities. DNA-binding specificities could be assigned with high confidence to nine heterodimers that displayed a range of DNA sequence preferences, including affinity for the CRE site preferred by the homodimer (*Figures 4A and B*). Importantly, distinct DNA-binding preferences among ATF3•CEBP and ATF3•BATF heterodimers and their corresponding homodimers were detected. The motifs enriched by the ATF3 homo- and heterodimers can be described in five broad categories: CRE, TRE, CRE-CAAT, CRE-L, and the emergent $^{5'}$TGACGCAT$^{3'}$ site (*Figure 4B*). Scatter plots illustrate instances where the CSI intensities of ATF3 heterodimers differ markedly from those of the parent homodimers (*Figure 4C and D*). For example, as evident from high CSI intensities, CRE-CAAT sites (red) are preferably bound by ATF3•CEBPG as compared to ATF3 or CEBPG (*Figure 4C*, top panel). Similarly, scores for TRE (green) and $^{5'}$TGACGCA$^{3'}$ (black) are higher for JUN•ATF3 than for JUN or ATF3 (*Figure 4C*, middle panel). BATF3•ATF3 (*Figure 4C*, bottom panel) and BATF2•ATF3 (*Figure 4D*, top panel) enrich CRE-L sites (blue), further supporting that CRE-L is an emergent site for BATF family heterodimers (also with JUN in *Figure 3C*). *Figure 4D* further highlights the differences between CRE and TRE binding by ATF3 in its homo versus heterodimer state. An important and recurring observation is that several ATF3 heterodimers (BATF3•ATF3, JUN•ATF3, and JUNB•ATF3) can associate with the CRE site that is bound by the ATF3 homodimer (*Figure 4*).

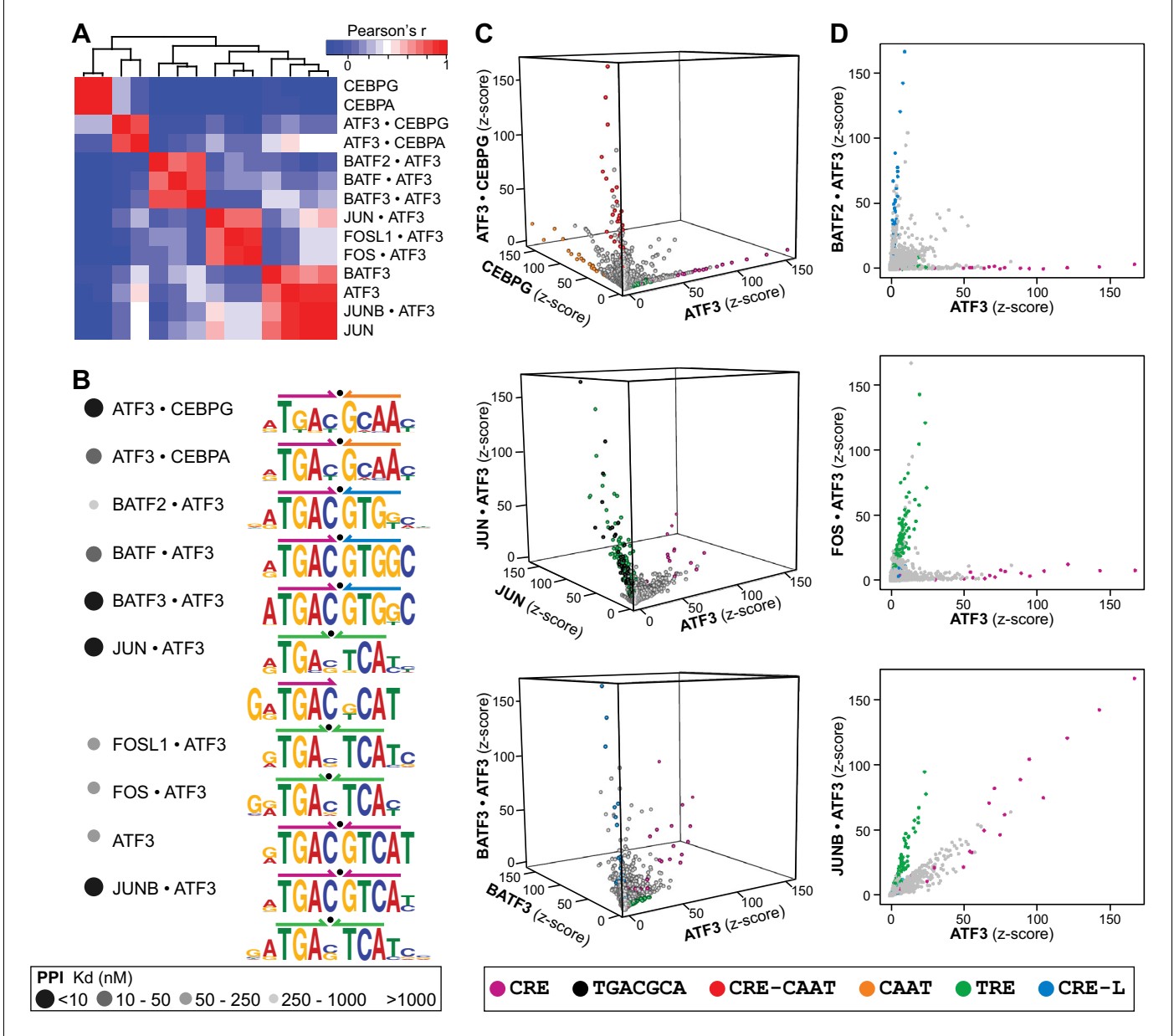

**Figure 4.** ATF3 heterodimers bind a range of distinct cognate sites. (**A**) Hierarchical clustering of pairwise comparisons of DNA-binding specificity (10-mers) for ATF3 homodimer and 9 ATF3-containing heterodimers. (**B**) DNA logos showing the MEME motifs derived from the top 1000 12-mer sequences for ATF3 homodimer and ATF3-containing heterodimers. Grey-scale circles next to dimer names indicate PPI strength using the scale from *Figure 3*. (**C**) 3-dimensional and (**D**) 2-dimensional scatter plots comparing the DNA-binding specificities of bZIP homodimers vs. ATF3-containing heterodimers. Scatter plots of quantile-normalized CSI intensities (z-scores) of ATF3 dimers for 80,000 10-mers are shown.

## Heterodimer sites enriched in SELEX-seq map to occupied genomic loci in vivo

To determine the extent to which cognate sites identified by SELEX-seq can explain genome-wide occupancy in cells, we examined ChIP-seq data for ATF3 in four different human cell lines. H1 human embryonic stem cells, HEPG2 liver-derived hepatocellular carcinoma cells, and K562 erythroblastoma cells have been examined comprehensively (*Encode, 2011*). The fourth cell line, GBM1 from Glioblastoma multiforme, is an aggressive brain cancer, wherein ATF3 is a tumor suppressor and its loss of function is indicative of high-grade cancer and poor prognosis (*Gargiulo et al., 2013*). As a first

step, we identified ATF3 ChIP-seq peaks and examined the overlap between the genomic sites occupied by ATF3 in all four lines. Only a small number of sites (119) were common between the four cell lines, although the number increased to 1602 genomic loci if only the ENCODE cells lines (H1, K562, and HEPG2) were examined (*Figure 5A*). This is a minor fraction of the over 10,000 peaks identified in K562 and about a third of the 4808 ATF3-bound sites in H1 cells.

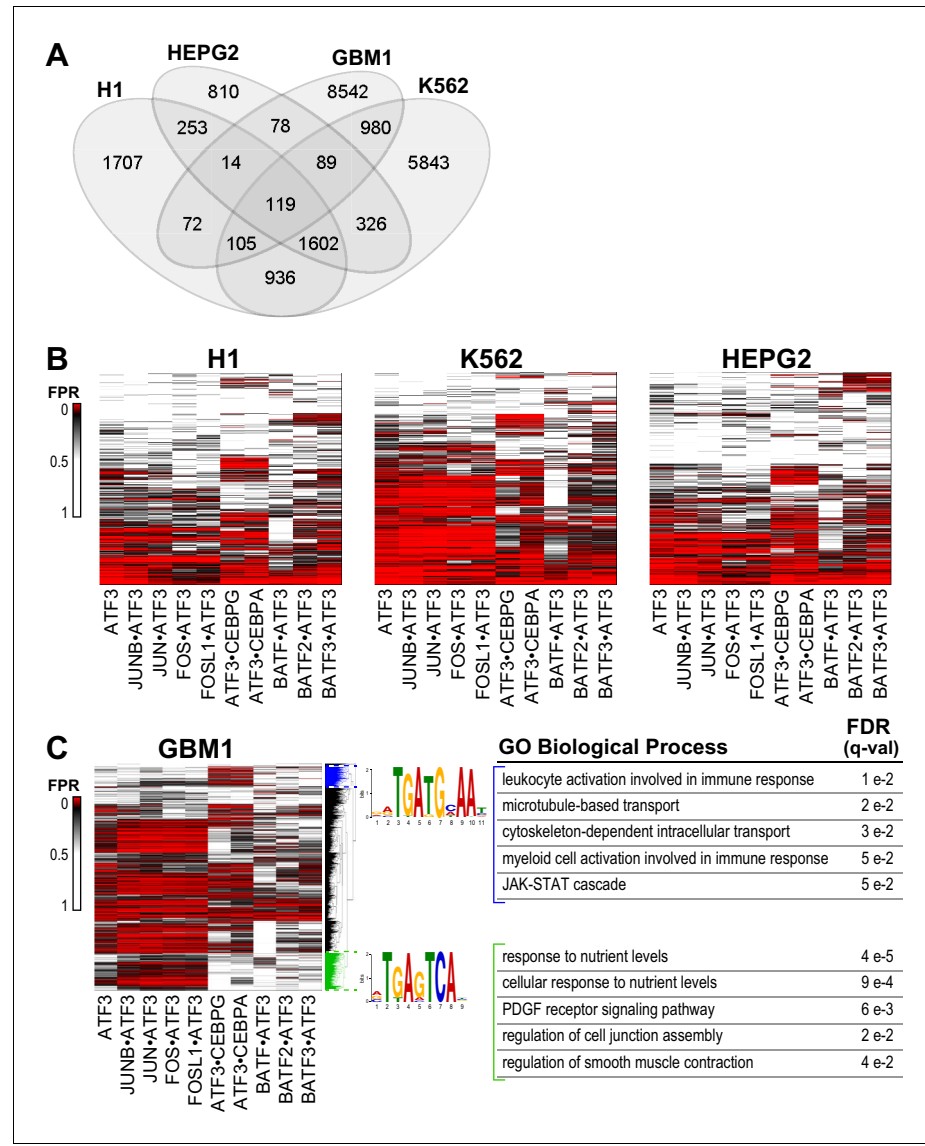

**Figure 5.** ATF3 binds to different genomic regions using diverse motifs. (**A**) Venn diagram of the numbers of ATF3-bound regions determined by ChIP-seq in different cell lines. (**B**) Heatmap of the False Positive Rate (FPR)-cutoffs at which ATF3 ChIP-seq peaks (rows) are detected as positive for ATF3 or ATF3-dimer binding. Peaks were scored using CSI intensities of the ATF3 homodimer or ATF3-containing heterodimers (columns) in H1hESCs, K562, and HEPG2 cells, and clustered by FPR-cutoffs across all dimers. (**C**) Same as (**B**) for the glioblastoma multiforme (GBM1) cell line. Highlighted clusters (blue and green) contain DNA motifs preferred by different ATF3 dimers and are enriched with different Gene Ontology Biological Process terms. False Discovery Rates (q-values) for each GO term are shown. See *Supplementary file 1H*.

The following figure supplement is available for figure 5:

**Figure supplement 1.** ROC curves.

ATF3 ChIP-seq peaks likely include both homodimer and heterodimer bound regions. To assess how well the in vitro discovered cognate sites explain bound sites in a cellular context, we used area under the curve-receiver operating characteristic (AUC-ROC) values, plotting the true-positive rate (TPR) versus false-positive rate (FPR) for peak detection (Materials and methods). ATF3 homodimer sites spanning the entire spectrum of CSI intensities (z-scores) yielded 0.67–0.77 AUC values (*Supplementary file 1E* and *Figure 5—figure supplement 1*). Using the AUC-ROC approach, we examined the ability of CSI profiles of nine different ATF3 heterodimers as well as the ATF3 homodimer to identify ATF3 ChIP-seq peaks that might represent heterodimer-bound regions. We used published RNA-seq datasets to verify the expression of the bZIP genes used for the ATF3 heterodimer analysis (*Supplementary file 1F*) (*Encode, 2011*; *Gargiulo et al., 2013*). Each of the 10 CSI datasets captures a large but varying fraction of the ATF3 peaks and, intriguingly, these data reveal that different ATF3 heterodimers perform better in different cell lines (*Supplementary file 1E*). For example, JUN•ATF3 gives 0.85 AUC in the Glioblastoma line, whereas BATF3•ATF3 better explains the ChIP-seq peaks in H1 and HepG2 cells with AUC of 0.69 and 0.70, respectively. While AUC-ROC curves are not robust to subtle changes, the differences we observe may reflect underlying cell line specific differences in the abundance and regulatory roles of different ATF3 heterodimers. The underlying epigenetic landscapes would further exacerbate these differences. Nevertheless, when considered together, the ATF3 homodimer combined with nine different heterodimers can account for a much larger fraction of ChIP-seq peaks than can the homodimer alone. For example, at an FPR-cutoff of 0.10, in the Glioblastoma cell line, the ATF3 homodimer classified just 39% of the ATF3 ChIP-seq peaks as positive, whereas 85% of the peaks are classified positive by at least one of the ATF3-containing dimers at FPR 0.10. Similar analysis for other cell lines and at different FPR cut-offs is reported in *Supplementary file 1G*.

Given the cell-type-specific differences in genomic sites occupied by ATF3, we scored the ATF3-bound loci for each cell line using the CSI data for 10 ATF3 dimers. Peaks were then clustered based on the FPR-cutoffs for each bound region (*Figure 5B–C* and Materials and methods). All four cell lines show clear clusters of sites where one or more heterodimer detects a peak at a lower FPR compared to the ATF3 homodimer. Several such clusters are apparent for heterodimers with CEBP or BATF family members. A striking result that emerged from the analysis of the GBM1 cell line is that multiple ATF3-bound genomic loci were better described by ATF3-heterodimers than the homodimer. For GBM1, we further examined two clusters of ChIP peaks for which heterodimers scored better than the ATF3 homodimer (*Figure 5C*, blue and green clusters in the dendrogram). In the blue cluster, de novo motif discovery revealed enrichment of a CRE-CAAT motif, which is the motif with maximal CSI intensities for CEBP•ATF3 dimers. De novo motif search of ChIP-seq peaks in the green cluster identified the TRE motif, which is the top ranked motif for ATF3 heterodimers formed with JUNB, JUN, FOS, and FOSL1, all of which are expressed in GBM1 cells (*Supplementary file 1F*). This is in contrast to the CRE motif preferred by the ATF3 homodimer. Gene ontology functional annotations of genes linked to the CRE-CAAT (blue) and TRE (green) clusters also differ substantially (*Figure 5C* and *Supplementary file 1H*). CRE-CAAT sites preferred by ATF3•CEBP heterodimers (blue cluster) enriched for gene ontology (GO) terms related to immune response and JAK-STAT signaling, whereas TRE sites (green cluster) enriched for GO terms associated to nutrient sensing, PDGF signaling, and cell junction regulation. This observation lends support to the notion that heterodimers drive cell-type and signal-specific gene networks.

## Co-occupied genomic loci bear emergent and conjoined sites

Sharpening our focus to a subset of genomic loci that are co-occupied by ATF3 and another bZIP permitted us to examine whether heterodimer cognate sites were evident at co-occupied genomic loci. In Tier 1 ENCODE cell lines such as H1 and K562, occupancy of multiple TFs has been charted across the genome (*Dunham et al., 2012*). We first examined loci co-occupied by ATF3 and CEBPB or JUN. In H1 embryonic stem cells, we identified a region on chromosome I that shows overlapping ChIP peaks for ATF3 and CEBPB (*Figure 6A*, top panel). This locus is also resistant to DNAse I, suggesting that ATF3 and CEBPB are binding to a seemingly inaccessible part of the genome. Plotting CSI intensities for a given TF across the genome generates CSI-Genomescapes (*Figure 6A–B*, bottom panels; Materials and methods). CSI-Genomescapes in the co-occupied region identified a high-intensity site for the ATF3•CEBPA heterodimer, whereas no high-intensity sequences were found for ATF3 or CEBPA homodimers (*Figure 6A*). CEBPA is the closest homolog to CEBPB for which CSI

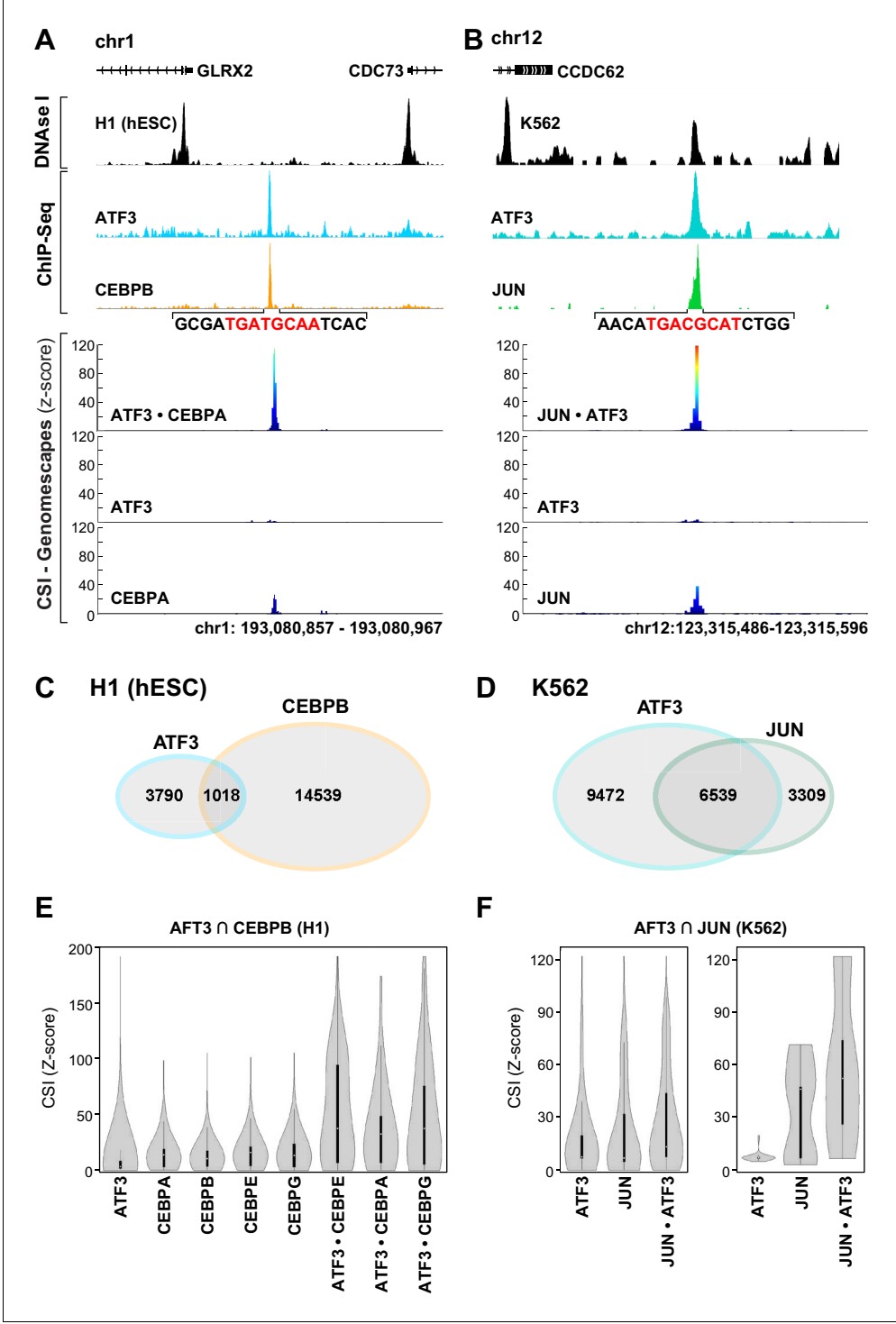

**Figure 6.** bZIP heterodimer DNA sites are bound in vivo. (**A**) ChIP-seq traces for ATF3 (blue) and CEBPB (orange) and DNase I hypersensitivity (black) trace for in H1 human embryonic stem cells. Below, CSI-Genomescape for bound genomic regions for ATF3 and CEBPA homodimers and ATF3•CEPBA heterodimer. CEBPA and CEBPB share 76% identity over their bZIP domain. (**B**) ChIP-seq traces for ATF3 (blue) and JUN (green) and DNAse I hypersensitivity trace (black) in K562 cells. Below, CSI-Genomescape for bound genomic region for ATF3 and JUN homodimers, and for JUN•ATF3 heterodimer. (**C**) Venn diagram of bound regions (ChIP-seq peaks) for ATF3 and CEBPB in H1hESC and for (**D**) ATF3 and JUN in K562 cells. (**E**) Violin plots of CSI-seq scores for the ChIP-seq peaks derived from the intersection of ATF3 and CEBPB ChIP peaks (1018 overlapping peaks) in H1 stem cells using in vitro data for ATF3, CEBPA, CEBPB (from Jolma et al.) (*Jolma et al., 2013*), CEBPE, CEBPG homodimers

*Figure 6 continued on next page*

*Figure 6 continued*

and ATF3•CEBPA, ATF3•CEBPE, and ATF3•CEBPG heterodimers. CSI intensities were quantile normalized. (**F**) Violin plots of CSI-seq scores for the ChIP-seq peaks derived from the intersection of ATF3 and JUN ChIP peaks (left, 6539 overlapping peaks) in K562 cells, left. Violin plots for the subset of overlapping peaks of ATF3 and JUN containing a match for the heterodimer-specific site TGACGCAT (39 peaks), right. Peaks were scored using ATF3 and JUN homodimers, and JUN•ATF3 heterodimers.

The following figure supplement is available for figure 6:

**Figure supplement 1.** CSI intensities for bound genomic regions.

data were obtained. Similar CSI-Genomescape analysis of a locus with overlapping ATF3 and JUN peaks readily identified the JUN•ATF3 emergent site ($^{5'}$TGACGCAT$^{3'}$). This site is within DNase I accessible euchromatin, and CSI-Genomescapes provide scant support for either JUN or ATF3 homodimer binding to this site (*Figure 6B*).

Next, we identified genomic loci that are co-occupied by ATF3 and CEBPB in H1 (1018 overlapping ChIP peaks) or ATF3 and JUN in K562 cells (6539 overlapping ChIP peaks; *Figure 6C–D*). We then used CSI data of different homo- and heterodimers to assign CSI scores within these co-occupied regions. Violin plots clearly demonstrate that regions co-occupied by ATF3 and CEBPB have higher CSI intensities when scored with ATF3•CEBP heterodimers than when scored with ATF3 and CEBP homodimers (*Figure 6E* and *Figure 6—figure supplement 1*). In contrast, for loci co-occupied by JUN and ATF3, violin plots indicate that cognate sites for JUN•ATF3 heterodimer perform only marginally better at explaining the genomic binding data than sites preferred by JUN or ATF3 homodimers (*Figure 6F*, left panel). This observation is consistent with the ability of the JUN•ATF3 heterodimer to bind consensus CRE sites that are also bound by each contributing homodimer. The perceptibly higher CSI intensity when using JUN•ATF3 cognate sites might arise from heterodimer-preferred TRE sites or heterodimer-specific emergent sites. To examine this possibility, we utilized CSI-Genomescapes to score all co-occupied regions that include emergent heterodimer-specific $^{5'}$TGACGCAT$^{3'}$ sites (39 sites). When this subset of genomic regions was examined with homodimer CSI data, the violin plots reveal the inability of ATF3 homodimer cognate sites to account for the ChIP signals, whereas JUN homodimers account for some of the JUN occupancy at those regions (*Figure 6F*, right panel). In contrast, the ATF3•JUN heterodimer cognate sites showed the highest scores for the emergent site.

## Heterodimer-specific cognate sites map to SNPs associated with diseases

Armed with 102 CSI profiles of bZIP dimers, we scrutinized 5076 non-coding single-nucleotide polymorphisms (SNPs) that are associated with diseases and quantitative traits (*Maurano et al., 2012*). We reasoned that non-coding SNPs that are not assigned to known TF cognate sites might be explained with our compendium of new bZIP-DNA interactomes. As a first step, we calibrated our CSI data by examining SNPs that are known to alter binding by CREB1 and CEBPA (*Figure 7A* top panel and *Figure 7—figure supplement 1A*). The minor allele of rs10993994 in the promoter of the MSMB gene has been associated with prostate cancer and it creates a cognate site that is bound by CREB1 (*Lou et al., 2009*). Similarly, the minor allele of rs12740374 has been associated with myocardial infarction, aberrant plasma levels of low-density lipoprotein cholesterol (LDL-C), and enhanced expression of SORT1 gene in the liver (*Musunuru et al., 2010*). Biochemical studies have demonstrated that the G-to-T change generates an optimal CAAT site that is bound by CEBPA. We applied CSI-Genomescape analysis to both SNPs. In both cases, the minor allele has a higher CSI intensity than the corresponding major allele, suggesting that the minor alleles of these SNPs create CEBPA- and CREB1-binding sites (*Figure 7A* and *Figure 7—figure supplement 1*). The CSI-Genomescape for rs7631605 site is particularly interesting because it predicts disruption of the emergent site $^{5'}$TGACGCAT$^{3'}$ (*Figure 7A* middle panel). This allele is associated to Alzheimer's disease and mild cognitive impairment (MCI) and elevated levels of phosphorylated Tau-181P (*Han et al., 2010*). Additionally, CSI-Genomescape predicts that rs1869901, a variant associated with schizophrenia, impacts binding of FOS•CEBPE by altering a TRE-CAAT site (*Figure 7A* bottom panel).

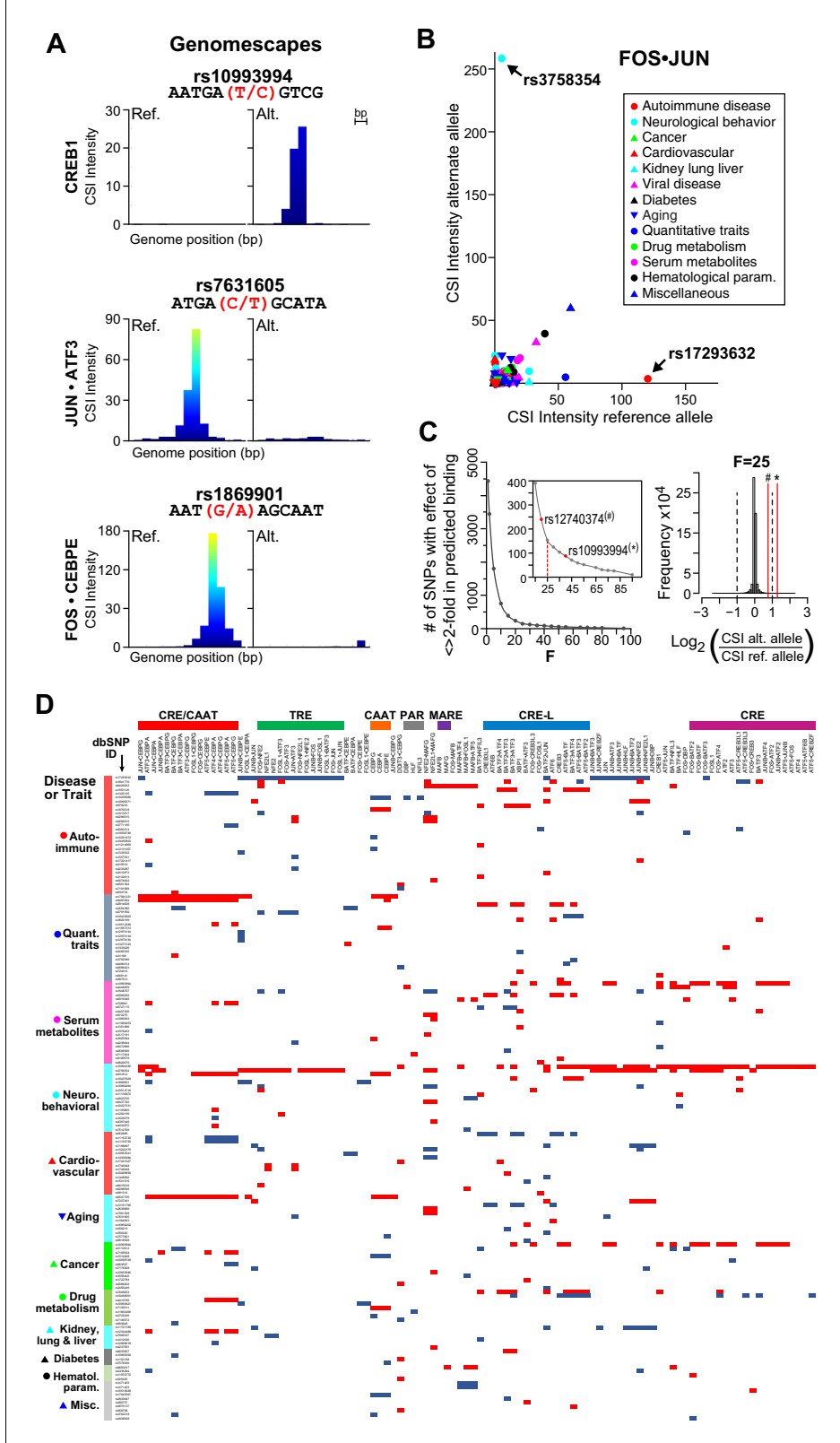

**Figure 7.** bZIP heterodimers and human diseases and traits. (**A**) CSI-Genomescape predicts increased binding by CREB1 to the alternate allele of rs10993994 and decreased binding to alternate alleles of rs7631605 and rs1869901 by JUN•ATF3 and FOS•CEBPE heterodimers, respectively. (**B**) Scatterplot of FOS•JUN predicted CSI intensities for reference and alternative alleles of 5076 autosomal SNPs linked to human diseases and quantitative traits

*Figure 7 continued on next page*

*Figure 7 continued*

identified in genome-wide association studies. SNPs and disease/traits classifications are from Maurano et al. (*Maurano et al., 2012*). (**C**) (left) Number of SNPs predicted to increase or decrease bZIP binding by twofold at different stringency levels determined by noise factor $F$ (see Materials and methods). The F values at which a twofold difference in CSI score is predicted for rs12740374 (#) and rs10993994 (*) are indicated in red. (right) Distribution of predicted fold changes in bZIP binding for GWAS SNPs using CSI Intensities, using $F = 25$. Dashed lines mark a twofold change. Red lines indicate the predicted change in binding of CREB1 and CEBPA to rs10993994 (*) and rs12740374 (#). (**D**) Predicted fold-change in CSI score of sequences centered at SNPs linked to disease or quantitative traits. A total of 156 SNPs have a predicted increase (red) or decrease (blue) of ≥2 fold in CSI score for at least one bZIP dimer, when $F = 25$ (Materials and methods and *Supplementary file 1I*). Fold-changes are relative to the reference genome hg19. Rows (SNPs) are organized by class of disease/trait. Columns (bZIP dimers) are clustered by DNA specificity as in *Figure 1*.

The following figure supplement is available for figure 7:

**Figure supplement 1.** Genomescapes, transcription factor binding, and chromatin environment for selected SNPs.

A scatterplot of CSI intensity scores for FOS•JUN (AP-1) for reference (hg19) or alternate alleles reveals SNPs that create or disrupt binding sites (*Figure 7B*). The plot shows that nearly all the 5076 SNPs are near the origin and do not lead to large differences in CSI scores for the FOS•JUN heterodimer. However, a striking example of predicted increase in binding is rs3758354, a SNP associated with schizophrenia, depression, and bipolar disorder (*Huang et al., 2010*). In contrast, a decrease in FOS•JUN heterodimer binding is predicted for rs17293632, a variant linked to Crohn's autoimmune disorder (*Franke et al., 2010*). ChIP-seq studies in several cell lines examined by the ENCODE consortium have shown binding by FOS and JUN to both loci, providing support that these sites are accessed by bZIP proteins in a cellular context (*Figure 7—figure supplement 1B*).

Extending beyond AP-1, we used 102 bZIP CSI profiles to score both alleles of the 5076 SNPs and calculated a predicted fold-change in CSI intensity, which correlates with binding affinity (*Figure 1—figure supplement 2*) (*Carlson et al., 2010*; *Puckett et al., 2007*). Similar correlations also hold true for other high-throughput platforms (*Berger et al., 2006*; *Fordyce et al., 2010*; *Slattery et al., 2011*). We added a noise factor to our scoring function to make the fold-change predictions less sensitive to low CSI intensities (*Figure 7C*; Materials and methods). A total of 156 SNPs yielded a greater than twofold difference in CSI intensity between the reference and alternate alleles (*Figure 7C–D*). Displaying the predicted increase (blue) or decrease (red) in binding by 102 bZIP dimers at 156 SNPs reveals minor alleles that are targeted by unique heterodimers as well as mutations that have wide-ranging impacts on multiple bZIP dimers. For example, rs10994336 is predicted to increase CSI intensity by at least twofold for 44 out of 102 bZIP pairs reported here. We also report that 80% of the identified changes impact bZIP heterodimers. In the richly annotated RegulomeDB database that ties SNP impact to occurrence of TF-binding sites, only 20 of 156 SNPs are currently annotated with a bZIP motif (*Boyle et al., 2012*). It is particularly important to note that many of the SNPs in the database are annotated with PWMs derived from bZIP homodimers, whereas our CSI intensity fold-change predictions for 22 homo- and 80 bZIP heterodimers make use of the entire bZIP-DNA interactomes (all 10-mers). The clusters in *Figure 7D* also point to potential roles of bZIP proteins in less understood diseases and provide new hypotheses for the etiology of such diseases and traits.

## Discussion

Transcription factors rarely function alone, different TFs are activated by different cellular stimuli, and specific combinations of TFs converge at specific genomic loci to regulate expression of genes (*Ptashne and Gann, 2002*). Such combinatorial control provides the means to integrate multiple signals and tune the expression of specific genes or sculpt genome-wide transcriptomes in a nuanced manner. The ability of different TFs to form hetero-oligomers via PPI and protein-DNA interaction is an essential feature of this process. While most eukaryotic TFs can bind DNA as monomers, the bZIP class of TFs only binds DNA as homo- or heterodimers. The ability of bZIPs to form heterodimers

appears to increase with increasing evolutionary complexity, with human bZIPs displaying more intricate heterodimerization networks than *C. elegans* and *D. melanogaster,* which in turn, exhibit more complex dimerization networks than *S. cerevisiae* (*Reinke et al., 2013*). Comprehensive PPI analysis has shown that 36 human bZIP monomers can form nearly 217 heterodimers, greatly expanding the repertoire of factors that can potentially bind DNA (*Reinke et al., 2013*). We demonstrate that this diversity of dimers expands the DNA sequence space that can be targeted by bZIPs. Our study further reveals that nearly 20% of the non-interacting bZIP pairs examined can be induced to dimerize at cognate DNA sites, providing yet greater diversity from a modest number of contributing monomers.

Given the large repertoire of human bZIP heterodimers, this family of TFs is particularly amenable to effect combinatorial control. Indeed, this potential was recognized long ago (*Bohmann et al., 1987*; *Franza et al., 1988*; *Lamb and McKnight, 1991*). An ever-increasing body of evidence now implicates bZIPs in numerous aspects of cellular and organismal function. Given their importance, a systematic study of DNA binding by bZIP heterodimers is clearly essential to understanding their functions. However, despite large surveys charting the TF-DNA interactomes (*Badis et al., 2009*, *2008*; *Berger et al., 2008*; *Carlson et al., 2010*; *Fordyce et al., 2010*; *Franco-Zorrilla et al., 2014*; *Grove et al., 2009*; *Jolma et al., 2010*, *2013*; *Kamesh et al., 2015*; *Nitta et al., 2015*; *Noyes et al., 2008*; *Siggers et al., 2012*; *Wei et al., 2010*; *Weirauch et al., 2014*), bZIP dimers were under-scrutinized with only a handful of heterodimers reported thus far (*Cohen et al., 2015*; *Jolma et al., 2015*; *Mann et al., 2013*). Thus, it was quite unclear prior to this work how dimerization between different bZIP partners would impact DNA recognition. The DNA-binding profiles for 80 heterodimers, which we report alongside equivalent data for 22 homodimers, is the largest bZIP heterodimer DNA-binding data reported to date and provides unprecedented insight into the impact of heterodimer formation.

Guided by PPI data, we examined the DNA-binding specificities of 126 stable dimers and 144 bZIP dimers that display no dimerization even at 1 μM. These 270 bZIP pairs represent a wide survey of the 666 potential pairs that can be formed by 36 monomers. The bZIP-DNA interactomes and specificity landscapes that emerged revealed three classes of cognate sites and several heterodimers displayed an ability to interact with more than one class of binding site. Of the three classes, conjoined half-sites were the most abundant, with nearly 72% of all heterodimers displaying some affinity for such sites. The second class contained variably-spaced half-sites, often overlapping by a single nucleotide. The final class, comprising 16% of the sites, was the least expected 'emergent' class of binding sites, where new non-obvious preferences for half-sites were revealed. Emergent sites targeted by heterodimers fall into 'loss of specificity' or 'gain of specificity' categories, as defined above. EMSA-FRET analyses not only quantified the relative affinities of hetero- and homodimers for these sites but also revealed the widespread ability of heterodimers to associate with cognate sites that have typically assumed to be bound by homodimers. More closely examining the emergent site targeted by ATF3•JUN, we find its occurrence at multiple locations across the human genome and, more importantly, several of these sites are co-occupied by ATF3 and JUN in vivo. Further emphasizing a physiological role for these non-obvious binding sites, a SNP that disrupts this site is linked to neurological diseases (*Han et al., 2010*).

The high granularity of our CSI data also revealed that sequences flanking well-studied homodimer motifs, such as CRE, can impart sub-structure to the motif that is recognized and preferentially bound by different bZIP pairs. Access to such nuanced specificity preferences allows better annotation of genome-wide binding data for bZIPs for which specificity profiles and high-quality ChIP data exist. This is particularly relevant because it is not uncommon for ChIP or genomic DNase I footprinting experiments to identify TF-bound regions that lack matches to the consensus motifs for a given TF. Our results suggest that a fraction of such in vivo occupied regions likely contain heterodimer binding sites. Another important insight from our comparative analysis of genome-wide binding profiles across four cell types is that a given heterodimer associates with distinct set of genomic loci in each cell type. The results suggest that underlying chromatin and epigenetic landscapes in different cell types may contribute significantly to the sites that are accessed by bZIP dimers. In this context, the ability of ATF3•CEBPB to bind a cognate site within seemingly closed chromatin is consistent with the ability of bZIPs such as CEBPB and FOS•JUN to function as 'pioneer' factors that first associate with closed chromatin and enable binding of additional TFs to yield transcriptionally active euchromatin (*Biddie et al., 2011*; *Garber et al., 2012*). Whether the ability to bind just one half site

is important, or whether DNA-templated dimerization of bZIPs confers any added ability to bind an otherwise inaccessible enhancer in closed chromatin, remains to be determined.

Finally, the specificity and binding energy profiles of 102 bZIP dimers enables a more nuanced examination of SNPs that have been linked by genome-wide association studies to various diseases and quantitative traits. The vast majority of SNPs associated with diseases occur in non-coding regions of the genome and most are not readily annotated by the available TF-DNA interactomes perhaps in part because the focus has been on obtaining consensus motifs of monomeric or homodimeric TFs. Rather than consensus motifs, the use of the full spectrum of binding specificity may enable more accurate mapping of TF-binding sites onto SNPs that are linked to diseases and phenotypic traits. Our compendium of CSI profiles accurately predicted creation of known bZIP cognate sites by previously validated SNPs. Of the 156 SNPs predicted by our CSI profiles to impact bZIP binding, nearly 77% were mapped to bZIP heterodimers, highlighting the importance of determining protein-DNA interactomes for heterodimer TFs. Nearly 64% created bZIP binding sites and were 'gain of function' changes relative to the human reference genome. These results are consistent with the 10-fold greater abundance of bZIP heterodimers over homodimers and the observation that aberrant stimulation of gene networks is arguably a greater contributor to disease etiology (*Bell et al., 2015*; *Lee and Young, 2013*; *Mansour et al., 2014*). SNPs that disrupt binding also contribute to disease, an example of this form of regulatory perturbation being the loss of the emergent JUN•ATF3-binding site that is associated with Alzheimer's and other neurological, cognitive and behavioral disorders. Our bZIP-DNA interactomes identify 156 SNPs that potentially impact 646 bZIP binding events, any one of these could potentially contribute to the associated ailments. Not only do our data help better annotate the genome, they also serve as an invaluable resource to generate hypotheses on how genetic variants may contribute to the etiology of a range of diseases.

The recent emergence of powerful high-throughput platforms for mapping protein-DNA interactomes has brought the goal of comprehensively mapping the binding specificities of all individual human TFs within reach (*Carlson et al., 2010*; *Jolma et al., 2013*; *Stormo and Zhao, 2010*; *Weirauch et al., 2014*). However, it is clear from our work as well as the recent work of others that the binding of TFs to each other, and/or to adjacent DNA sites, can influence binding specificity profiles in important ways (*Ansari and Peterson-Kaufman, 2011*; *Garvie et al., 2001*; *Grove et al., 2009*; *Jolma et al., 2015*; *Mann et al., 2013*; *Siggers et al., 2012*; *Slattery et al., 2011*). Our study heralds the important next wave of specificity mapping, in which the field will tackle the effects of higher order interactions and begin to relate these to the transcriptional control of key biological processes.

# Materials and methods

## bZIP cloning, expression, and labeling

Human bZIP proteins containing the basic-region and coiled-coiled domains with an N-terminal 6x His tag and a C-terminal intein-chitin binding domain were expressed as described previously (*Reinke et al., 2013*). Sequences are in *Supplementary file 1A*. Briefly, *Escherichia coli* RP3098 cells transformed with bZIP clones were grown in 0.5 L LB cultures at 37°C to OD600 = 0.4–0.8. Expression was induced with the addition of 0.5 mM IPTG (Isopropyl $\beta$-D-thiogalactopyranoside) and cultures incubated for 3–4 hr at which point cells were pelleted. Cells pellets were resuspended in 20 mM HEPES pH 8.0, 500 mM NaCl, 2 mM EDTA (ethylenediaminetetraacetic acid), 1 M guanidine-HCl, 0.2 mM PMSF (phenylmethylsulfonyl fluoride), and 0.1% Trition X-100). Cells were then sonicated and the lysate poured over a column of 1 ml chitin beads to bind the protein (NEB, Ipswich, MA). The column was then washed and equilibrated in EPL buffer (50 mM HEPES pH 8.0, 500 mM NaCl, 200 mM MESNA (2-mercaptoethanesulfonic acid), 1 M guanidine-HCl). The bZIP domain was then cleaved from the intein and labeled with biotin on the C-terminus by incubation for at least 16 hr in 1 ml EPL buffer containing 1 mg/ml cysteine-lysine-biotin peptide (CELTEK Peptides, Nashville, TN). The cleaved and biotin-labeled proteins were then eluted from the column using EPL buffer without MESNA and then diluted fivefold into denaturing buffer (6 M guanidine-HCl, 5 mM imidazole, 0.5 M NaCl, 20 mM TRIS, 1 mM (DTT) Dithiothreitol, pH 7.9) and bound to a column containing 1 ML Ni-NTA beads (QIAGEN, Hilden, Germany). Columns were washed and proteins eluted with 60% ACN (Acetonitrile)/0.1% TFA (Trifluoroacetic acid). The labeled proteins were then lyophilized,

resuspended, and desalted using spin-columns (Bio-Rad, Hercules, CA). Proteins were stored in 10 mM potassium phosphate pH 4.5 at −80°C. Peptide concentrations were determined by measuring absorbance at 280 nM in 6 M guanidine-HCl/100 mM sodium phosphate pH 7.4. The fluorescein and TAMRA-labeled proteins used in gel-shift assays were generated as described previously (*Reinke et al., 2013*).

## Cognate site identification by HT-SELEX

Cognate-binding sites for bZIP homo- and heterodimers were determined by SELEX-seq (*Jolma et al., 2010*; *Tietjen et al., 2011*; *Zhao et al., 2009*; *Zykovich et al., 2009*). A DNA library (Integrated DNA Technologies, Inc.) with a central randomized 20 bp region ($10^{12}$ possible sequences), flanked by constant sequences used for amplification was used (*Supplementary file 1B*). In vitro selections were performed as follows. For bZIP homodimers, purified, C-terminal biotinylated-bZIP proteins (50 nM) were added to 100 nM of DNA library (Binding buffer: 1x PBS (10 mM $PO_4^{3-}$, 137 mM NaCl, 2.7 mM KCl), pH 7.6, 2.5 mM DTT, 50 ng/μl poly dI-dC, 0.1% BSA) and incubated at room temperature for 1 hr. The DNA library concentration and volume (20 μl) were such that there was a high probability of sampling at least one copy of every 20-mer sequence ($10^{12}$ permutations). bZIP-DNA complexes were enriched with streptavidin-coated magnetic beads (Dynabeads, Invitrogen, Carlsbad, CA) following the manufacturer's protocol. After pull-down, three quick washes with 100 μl ice-cold binding buffer were performed to remove unbound DNA. Beads were resuspended in a PCR master mix (EconoTaq PLUS 2X Master Mix, Lucigen) and the DNA was amplified for 15 cycles. Amplified DNA was column purified (QIAGEN), quantified by absorbance at 260 nm, and used for subsequent binding rounds. Three rounds of selection were performed. For bZIP heterodimers, one bZIP partner had a C-terminal biotin tag. bZIP-DNA complexes were pulled down with streptavidin-coated magnetic beads. Several steps were followed to decrease DNA binding by competing homodimers: (1) a 1:3 molar ratio with an excess of the non-biotinylated bZIP was used to shift the thermodynamic equilibrium from the biotin-labeled homodimer; (2) the biotin-bZIP used for pull-down was chosen as the more weakly interacting homodimer of the two interaction partners. As a convention, when naming bZIP heterodimers, the bZIP-biotin is listed first, unless otherwise stated. After three rounds of selection, an additional PCR was done to incorporate Illumina sequencing adapters and a unique 6 bp barcode for multiplexing. The starting library (Round 0) was also barcoded. Up to 180 samples were combined and sequenced in a single Illumina GAIIx or HiSeq2000 lane.

## Sequencing data analysis

Illumina sequencing yielded ~180 million reads per lane. Reads were de-multiplexed by requiring an exact match to the 6 bp barcode and truncated to include only the 20 bp derived from the random portion of the library. On average, we obtained 850,000 reads per barcode. The occurrence of every k-mer (lengths 8 through 14 bp) was counted using a sliding window of size k. To correct for biases in our starting DNA library, we took the ratio of the counts of every k-mer to the expected number of counts in the starting library. The starting library was modeled using a fifth-order Markov Model derived from the sequencing reads corresponding to the starting library (Round 0) (*Slattery et al., 2011*). We then calculated a CSI intensity (z-score = $(x - \mu)/ \sigma$) for each k-mer, using the distribution of k-mer enrichment values for that dimer. The most enriched 10, 12, and 14 bp subsequences were used to derive PWM motifs using MEME. Samples that failed to enrich specific sequences relative to the starting library (Round 0) or that only enriched low-complexity sequences were not included in further analysis. Data files for 20 bp reads and normalized 10 bp sequences are available at https://ansarilab.biochem.wisc.edu/computation.html.

Previously reported bZIP-DNA interaction data were downloaded from study PRJEB3289 in the European Nucleotide Archive (http://www.ebi.ac.uk/ena/data/view/PRJEB3289) (*Jolma et al., 2013*). 20 bp reads for bZIP proteins and their corresponding 20 bp DNA library (round 0) were analyzed as described previously.

## Homodimer and heterodimer clustering

Binding profiles were defined for each bZIP pair using the CSI intensities (z-scores) of 1222 unique 10-mer sequences. This set of 10-mers is composed of the 50 highest-scoring sequences for each

dimer. Unsupervised hierarchical clustering of pair-wise binding profile similarities, assessed by Pearson's correlation coefficient r, was done using R. Dendrograms and heatmaps were generated using the *heatmap.2* function in the gplots R-package. Heterodimers were labeled as such if the bZIP-DNA complex was pulled-down by a biotinylated bZIP that does not binds DNA as a homodimer in our experimental conditions. If the bZIP used for pull-down of the bZIP heterodimer also bound DNA as a homodimer, the observed DNA specificity was assigned to the heterodimer only if the heterodimer specificity landscape was different (t-test $p<0.05$) from the homodimer specificity, assessed by correlation scores (*Figure 1—figure supplement 4*).

## Sequence logos

PWMs were derived from the 1000 most enriched 12-mer sequences (ranked by z-score) for each bZIP pair, using the MEME (*Bailey and Elkan, 1994*). The most enriched 14-mer sequences were used for MAF dimers. MEME was run with following parameters: -dna -mod anr -nmotifs 10 -minw 8 -maxw 18 -time 7200 -maxsize 60000 –revcomp.

## Specificity and energy landscapes

SELs display high-throughput protein-DNA (or protein-RNA) binding data for both array and sequencing methods (*Campbell et al., 2012*; *Carlson et al., 2010*; *Tietjen et al., 2011*). The organization of data in SEL is detailed in *Figure 2A*. The SELs shown in this work were generated from 10-, 12-, or 14-mer intensity files. Seed motifs were derived from PWM-derived DNA logos or from the highest intensity k-mer, and are shown on top of each SEL. The length of the seed motifs has to be smaller than the k-mer length of the CSI intensity file. The software to generate SELs is provided as Source Code Files (SEL_10MER and SEL12MER_14MER).

## Electrophoretic mobility shift assay–FRET

An electrophoretic mobility shift assay (EMSA) with fluorescence resonance energy transfer (FRET) readout was used to validate bZIP heterodimer binding to DNA. The assay relies on the ability to observe FRET between two fluorophores, TAMRA and fluorescein, as well as to detect each fluorophore in the absence of FRET (*Figure 3A*). The assay also measures DNA fluorescence to ensure that protein-DNA complexes are being examined. Two versions of each bZIP were made, one conjugated to TAMRA and the other to fluorescein. We observed that the fluorophores reproducibly retard (TAMRA) or increase (fluorescein) the mobility of the bZIP protein that they are attached to and thus assist in resolving each heterodimer with respect to the two homodimers formed by contributing partners. The sequences of all the DNA sites used are listed in *Supplementary file 1D*. Each site was flanked by six constant nucleotides on each side (GAGTCC-site-CCGTAG). Oligos modified on the 5′ end with the dye TYE 665 (IDT, Coralville, IA) were annealed with an unlabeled reverse-complement oligo. Binding reactions contained 50 nM of each fluorescein- and TAMRA-labeled proteins, 10 nM annealed dye-labeled DNA in 20 µl of binding buffer (50 mM potassium phosphate pH 7.4, 150 mM KCl, 0.1% BSA, 0.1% Tween-20, 5 ng/µl poly (dI-dC), 0.5 mM TCEP). Samples were mixed, incubated at 37°C for 30 min, and then at 21°C for 30 min. NOVEX 6% DNA retardation gels were loaded with 16 µl of each sample (Life Technologies, Carlsbad, CA) and run at 300V for 20–22 min at 22–25°C. Gels were then imaged using a Typhoon 9500 scanner (GE Healthcare Bio-Sciences Corp., Piscataway, NJ) with separate channels for fluorescein, TAMRA, TYE 665, and FRET. Bleed through between channels was corrected using the spectral-unmixing plugin in ImageJ (http://rsb.info.nih.gov/ij/). The amount of DNA bound for the homodimers was calculated by quantifying the DNA signal that corresponded to all three bound species (fluorescein homodimer, TAMRA homodimer, and mixed-dye homodimer). For the heterodimers, the amount of DNA bound was calculated by quantifying the DNA signal that corresponded to the mixed dye heterodimer. The amount of bound DNA was divided by the amount of unbound DNA run without protein added. For each heterodimer, the interaction was measured twice, with the fluorescein and TAMRA dye on different proteins, and the average of the two measurements is reported.

## ChIP-seq data

ChIP-seq peaks from the ENCODE project used in this work were downloaded from ftp://hgdownload.cse.ucsc.edu/goldenPath/hg19/encodeDCC/wgEncodeAwgTfbsUniform/ (*Dunham et al.,*

*2012*). Overlapping genomic regions of ChIP-seq peaks were determined and extracted using bed-ops (*Neph et al., 2012*). For ATF3 ChIP-seq in GBM1 cells, aligned reads (.bam file) were downloaded from GEO (GSE33912). ATF3 peaks were called using the MACS tool (*Zhang et al., 2008*) in the Galaxy (*Goecks et al., 2010*) platform using default parameters. Overlapping ATF3-bound regions between different cell lines (*Figure 5*) were determined using the ChIPpeakAnno R-package (*Zhu et al., 2010*).

## CSI genomescapes: scoring in vivo bound sites with in vitro data

A CSI Genomescape is a plot generated by assigning in vitro CSI intensities (z-scores) to genomic regions. To generate the CSI Genomescapes in *Figures 6* and *7*, a 10 bp sliding window was used to score reported ChIP-seq peaks using quantile-normalized CSI intensities for different bZIP dimers as follows: Given a bZIP pair and a ChIP-seq peak, the peak was assigned the maximum CSI intensity for any 10-mer within the reported peak.

## Receiver operating characteristic

CSI Genomescapes of ChIP-seq data sets were then used to generate Receiver Operating Characteristic (ROC) curves to reflect how well the in vitro binding data for different bZIPs explains the ChIP-seq data. In this analysis, ChIP-seq peaks were used as a true positive set, whereas two regions of equal length ±5 kbp from the center of each peak (that did not overlap another ChIP-seq peak) were chosen to make the true negative set. The fraction of regions in the positive vs. negative sets with scores above a varying CSI intensity cutoff were plotted to generate ROC curves (True Positive Rate vs. False Positive Rate). ATF3-bound regions (ChIP-seq peaks) were scored with the CSI intensities for ATF3 homodimer or for ATF3-containing heterodimers to generate the areas under the curves. Heatmaps and clustergrams in *Figure 5* were made by hierarchical clustering of the lowest FPR-cutoff values at which peaks were detected as positives using the CSI intensities of the ATF3 containing dimers. ROC curves and heatmaps were generated in MATLAB.

## De novo motifs and functional annotation of ChIP-seq peaks

Motif finding within ChIP-seq peaks was done with MEME-ChIP with default settings (*Machanick and Bailey, 2011*). Enrichment of functional annotations of genomic regions was done with Genomic Regions Enrichment of Annotations Tool (GREAT) with default settings (*McLean et al., 2010*). Gene Ontology annotations that are significantly enriched (FDR < 0.05) by both binomial and hypergeometric test are shown. The False Discovery Rate (q-value) is corrected for multiple hypothesis tests.

## Single-nucleotide polymorphism scoring

SNPs linked to diseases or quantitative traits by GWAS were obtained from the Supplemental Table S2 from Maurano et al., which reports human SNPs associated to diseases and quantitative traits (*Maurano et al., 2012*). For each SNP, we considered 21 bp region centered on the SNP (10 bp on each side) and assigned a score using the CSI intensity data all 10-mers. We scored both alleles using a 10 bp sliding window and assigning the highest CSI intensity (z-score) in the 21 bp fragment; each 21 bp region was scored with twelve 10 bp windows. We calculated a predicted fold-difference in CSI intensity between a given SNP and its reference allele (hg19) using the following formula:

$$\frac{(\text{CSI Intensity for alternate allele} - \text{Minimum CSI Intensity} + A)}{(\text{CSI Intensity for reference allele (hg19)} - \text{Minimum CSI Intensity} + A)}$$

where the $A$ = (Maximum CSI Intensity – Minimum CSI Intensity) * $F$, Minimum CSI Intensity = minimum CSI Intensity (z-score) among the scored SNPs, Maximum CSI Intensity = maximum CSI Intensity (z-score) among the scored SNPs. And $F$ is a noise factor which was varied from 1% to 90%, from lower to higher stringency in estimating the predicted difference in CSI intensity. We added a noise factor (F) to the formula to make the fold-change prediction less sensitive to low CSI scores and decrease the number of false-positives predictions.

## Acknowledgements

We thank Professor Parmesh Ramanathan and members of the Ansari and Keating laboratories for helpful discussions, Christos Kougentakis for technical assistance with EMSA assays, Laura Vanderploeg for help with the artwork, and Marie Adams from the University of Wisconsin Biotechnology Center DNA Sequencing Facility. This study was supported by NIH award R01 GM096466 to AEK, and NIH grants R01 CA133508 and U01 HL099773, and the W M Keck Medical Research Award to AZA. JARM was supported by the National Human Genome Research Institute (NHGRI) training grant of the Genome Sciences Training program (T32 HG002760).

## Additional information

### Competing interests

AZA: The sole member of VistaMotif, LLC and founder of the nonprofit WINStep Forward. The other authors declare that no competing interests exist.

### Funding

| Funder | Grant reference number | Author |
| --- | --- | --- |
| National Institutes of Health | R01GM096466 | Amy E Keating |
| W. M. Keck Foundation | | Aseem Z Ansari |
| National Institutes of Health | U01HL099773 | Aseem Z Ansari |
| National Institutes of Health | R01CA133508 | Aseem Z Ansari |
| National Institutes of Health | T32HG002760 | Jose A Rodriguez-Martinez |

The funders had no role in study design, data collection and interpretation, or the decision to submit the work for publication.

### Author contributions

JAR-M, AWR, Conceptualization, Formal analysis, Investigation, Methodology, Writing—original draft, Writing—review and editing; DB, Conceptualization, Software, Formal analysis, Investigation, Methodology, Writing—original draft, Writing—review and editing; AEK, AZA, Conceptualization, Funding acquisition, Methodology, Writing—original draft, Writing—review and editing

### Author ORCIDs

José A Rodríguez-Martínez, http://orcid.org/0000-0002-1191-2887
Aaron W Reinke, http://orcid.org/0000-0001-7612-5342
Aseem Z Ansari, http://orcid.org/0000-0003-1432-4498

## Additional files

### Supplementary files

• Supplementary file 1. (A) bZIP sequences. (B) DNA library and primers. (C) DNA stabilized bZIP dimers. (D) Oligonucleotide sequences for EMSA. (E) ROC-AUC. (F) Expression of bZIP genes. (G) ATF3 dimers in ChIP-seq peaks. (H) GREAT GO annotations. (I) SNP fold-change predictions.

• Supplementary file 2. MEME motifs and Sequence Specificity and Energy Landscapes (SEL) for human bZIP homodimers and heterodimers.

### Major datasets

The following dataset was generated:

**Database, license, and accessibility**

| Author(s) | Year | Dataset title | Dataset URL | information |
|---|---|---|---|---|
| José A Rodríguez-Martínez, Aaron W Reinke, Devesh Bhimsaria, Amy E Keating, Aseem Z Ansari | 2017 | Data from: Combinatorial dimerization of human bZIP factors confers preferences for different classes of DNA binding sites | https://ansarilab.bio-chem.wisc.edu/computa-tion.html | Publicly available on the Ansari Lab (https://ansarilab.biochem.wisc.edu/) |

The following previously published datasets were used:

| Author(s) | Year | Dataset title | Dataset URL | Database, license, and accessibility information |
|---|---|---|---|---|
| Gargiulo G, Cesaroni M, Serresi M, Lancini C, De Vries N, Hulsman D, van Lohuizen M | 2013 | Functional Identification of Critical Bmi1 target genes in Neural Progenitor and Malignant Glioma cells | http://www.ncbi.nlm.nih.gov/geo/query/acc.cgi?acc=GSE33912 | Publicly available at the NCBI Gene Expression Omnibus (accession no: GSE33912) |
| Jolma A, Yan J, Whitington T, Toivonen J, Nitta KR, Rastas P, Morgunova E, Enge M, Taipale M, Wei G, Palin K, Vaquerizas JM, Vincentelli R, Luscombe NM, Hughes TR, Lemaire P, Ukkonen E, Kivioja T, Taipale J | 2017 | DNA-binding specificities of human transcription factors | http://www.ebi.ac.uk/ena/data/view/PRJEB3289 | Publicly available at the EMBL European Nucleotide Archive (accession no: PRJEB3289) |

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
