## [Decision Letter]

Thank you for submitting your article "Combinatorial bZIP dimers define complex DNA-binding specificity landscapes" for consideration by *eLife*. Your article has been reviewed by two peer reviewers, including Michael Green who is a member of our Board of Reviewing Editors, and Kevin Struhl (who is not responsible for the references to previous work in the reviews) as the Senior Editor.

The reviewers have discussed the reviews with one another and the Reviewing Editor has drafted this decision to help you prepare a revised submission.

Summary:

This is an interesting manuscript on the DNA binding specificity of dimers in the human bZIP family. bZIP proteins are known to play critical regulatory roles in many cellular processes, and different members of this family are known to form both homo- and heterodimers (sometimes with more than one partner). Prior to the work described in this manuscript, it was unclear how dimerization influences DNA specificity. The authors used SELEX-seq experiments to characterize the binding specificities of 80 heterodimers and 22 homodimers, and used proper control experiments to ensure that the data is interpreted correctly (e.g. they tested that the choice of which bZIP monomer is biotinylated does not affect results).

This rich data set has the potential to change the way we analyze genomic binding of bZIP proteins, especially in terms of assessing the potential impact of SNPs on bZIP-DNA binding. However, in its current form, the manuscript is not ready for publication. Several sections are unclear and hard to read, and there are many incorrect references to figures or supplementary material. Please see specific comments and questions below.

Essential revisions:

1) The identification of "emergent sites" for some bZIP dimers is one of the major insights gained from the authors' large compendium of bZIP specificities. According to the authors, such sites cannot be predicted from the specificities of the homodimers. However, the section presenting this result, "Conjoined, Variably-spaced and Emergent cognate sites bound by heterodimers" is very hard to read.

The supplementary files are not numbered correctly: "[…] the homodimer DNA specificity (e.g., ATF4 vs. ATF4•CEBPA, r = 0.1; Figure 1—figure supplement 3)." I could not find this figure.

In Figure 1, please annotate in the figure itself which motifs are "conjoined" vs. "variably-spaced" vs. "emergent". Arranging the motifs into these categories would make the manuscript easier to follow. For the motifs that are compared to one another, the authors should display them together and aligned them. Otherwise it is very hard to follow what the authors are trying to convey (see subsection “Conjoined, Variably-spaced and Emergent cognate sites bound by heterodimers” paragraph two). A direct comparison of FOS-CABPG and FOS-CABPE should be shown in a figure.

Related to the same paragraph and Figure 1, it is interesting that BATF3 as a homodimer binds TGACGTCA, but as heterodimers with CEBPA, ATF3, ATF5, and NFIL3, BATF proteins bind GTGG (a CRE-L half-site). Could it be that as homodimers BATF proteins can bind both CRE and CRE-L sites, but these two different sites cannot be captured by a single motif? Certain bZIP subfamilies have been known for many years to bind more than 1 type of motif. For example, see early work from Kevin Struhl's lab on Ap-1 versus ATF/CREB bZIP domains: Ap-1 proteins recognize overlapping TGAC half-sites TGA(C|G)TCA, while ATF/CREB proteins recognize adjacent half-sites TGACGTCA; however, "AP-1 proteins prefer to bind AP-1 sites, but they also bind ATF/CREB sites with only slightly lower affinity" (PMID: 7630732). Given that the authors use single motifs to represent the specificity of homo and heterodimers, such preferences for more than one arrangement of half-sites is not illustrated in the paper.

2) From the manuscript it seems clear that if we take only the top motif for each dimer, then "emergent" motifs cannot be predicted from the motifs of the individual homodimers. But if for each homodimer we consider all the motifs they recognize (sometimes with different affinities, other times with very similar affinity), can we then predict some of the "emergent" motifs?

On a related note: The authors mention that "Nine out of the 80 heterodimers (11%) enriched two motifs ([Supplementary-material SD4-data]). For example, BATF•CEBPG enriched both CRE-CAAT and CRE-L-CAAT motifs." But what about homodimers? Some homodimers might also bind two motifs.

The authors tried to address this issue by using "specificity and energy landscapes" (SEL). While I understand that these landscapes contain a lot more information than the motifs, they are not intuitive to interpret and it is not clear to me how one would immediately extract information about multiple half-site arrangements from SELs. A much simpler representation would be by using several motifs for each homo/hetero-dimer that binds more than 1 half-site arrangement.

3) Comparing SELs is also difficult. When two dimers have a site in common, how can we see that from the SELs? Unless the seed is exactly the same, the SELs are hard to compare.

The authors interpret some SELs in subsection “Specificity and Energy Landscapes (SELs) reveal the entire spectrum of cognate sites bound by heterodimers”, but couldn't the same insights be gained by simply deriving more than 1 motif from each data set?

Maybe one possibility for presenting the data is to generate a set of motifs representing all the half-site combinations observed in the data, and then for each dimer plot its affinities for sites matching each motif.

4) Few studies of heterodimers binding to DNA carefully dissect the binding of hetero- versus homodimers. The authors did this using EMSA-FRET assays, and for a large number of heterodimers (67).

Although this data is very valuable, the way it is presented is not clear – see Figure 3 and associated text. I would recommend using a single color for PPIs, and a single color for PDIs (with intensity proportional to affinity of the interactions). Or even have separate plots for PPIs versus PDIs. Alternatively, the authors could show the data as barplots with PDI strength above the x-axis (i.e. as positive numbers) and PPIs underneath the x-axis (i.e. as negative numbers). Another option would be to use circles where the size reflects PDI strength and the color intensity reflects PDI strength. In subsection “EMSA-FRET analysis to validate heterodimer binding to different cognate sites” the authors say: "It is readily evident that neither the JUN nor the ATF3 homodimers associate with the emergent site identified by SELs for JUN•ATF3 (Figure 3 and Figure 3—figure supplement 1)." To this reviewer it was not evident, and it took quite a lot of time to extract this information from SELs and the figures in the manuscript.

5) The case study of ATF3's specificity being influenced by partner TFs is clear and interesting.

6) Regarding the in vivo ATF3 analysis: the authors found that the motifs of different ATF3 dimes are best at explaining the ATF3 ChIP-seq data in different cell lines. Did the authors verify that the partners identified by their analysis are actually expressed, at high enough levels, in those cell lines?

Also, are the reported differences in AUC-ROC values between different motifs (Figure 5) significant? What is the magnitude of differences observed for motifs trained on replicate SELEX-seq experiments? The AUC-ROC values are generally not robust to small changes in the motifs. The analysis shown in Figure 5, based on the SELEX-seq data, better reflects the cell-type specific differences in ATF3 heterodimer binding.

7) The analysis in the last section of Results (Figure 7) is very interesting. However, the "noise factor" used in the analyses of SNP data seems ad-hoc. What is the rationale for the formula used to calculate the noise factor?

---

## [Author Response]

*Essential revisions:*

*1) The identification of "emergent sites" for some bZIP dimers is one of the major insights gained from the authors' large compendium of bZIP specificities. According to the authors, such sites cannot be predicted from the specificities of the homodimers. However, the section presenting this result, "Conjoined, Variably-spaced and Emergent cognate sites bound by heterodimers" is very hard to read.*

We have revised this section of the manuscript extensively and worked to make it more accessible. We now present the reasoning that led us to describe heterodimer-preferred sites as emergent sites (Results section “Conjoined, Variably-spaced, and Emergent cognate sites bound by heterodimers”, third paragraph). We hope the clarity of this section is now improved.

*The supplementary files are not numbered correctly: "[…] the homodimer DNA specificity (e.g., ATF4 vs. ATF4•CEBPA, r = 0.1; Figure 1—figure supplement 3)." I could not find this figure.*

We have corrected figure supplement numbers in the revised manuscript.

*In Figure 1, please annotate in the figure itself which motifs are "conjoined" vs. "variably-spaced" vs. "emergent". Arranging the motifs into these categories would make the manuscript easier to follow. For the motifs that are compared to one another, the authors should display them together and aligned them. Otherwise it is very hard to follow what the authors are trying to convey (see subsection “Conjoined, Variably-spaced and Emergent cognate sites bound by heterodimers” paragraph two). A direct comparison of FOS-CABPG and FOS-CABPE should be shown in a figure.*

We have rearranged the representative heterodimer motifs that are shown in Figure 1. Motifs are now grouped and labeled as conjoined, variably spaced, or emergent. As suggested by the reviewer, FOS-CEBPG (motif 1) is now aligned and placed on top of FOS-CEBPE (motif 4) to facilitate the comparison. In addition, we now include direct comparison of FOS•CEBPG vs. FOS•CEBPE and FOSL1•CEBPE vs FOSL1•CEBPG in Figure 1—figure supplement 5.

*Related to the same paragraph and Figure 1, it is interesting that BATF3 as a homodimer binds TGACGTCA, but as heterodimers with CEBPA, ATF3, ATF5, and NFIL3, BATF proteins bind GTGG (a CRE-L half-site). Could it be that as homodimers BATF proteins can bind both CRE and CRE-L sites, but these two different sites cannot be captured by a single motif?*

The sequence preferences of BATF, BATF2 and BATF3 homodimers were examined in this study, and of these, only BATF3 homodimer yielded a motif in our SELEX-seq studies (see [Supplementary-material SD3-data] for the complete list of bZIP dimers examined). The data in Figure 1, Figure 2, and 3C and in [Supplementary-material SD4-data] (logos and landscapes) indicate that BATF3 homodimer binds to the CRE site. The CRE-L motif was not identified using the MEME motif-finding algorithm, and EMSA-FRET analysis showed that BATF3 homodimer binds to CRE, but not to CRE-L (Figure 3). We conclude that CRE-L binding by BATF3 is extremely weak, if it occurs at all.

Related to the MEME search, we agree with the reviewer that a single motif would not capture both CRE and CRE-L sites. However, we respectfully note that our motif searches were not restricted to single motifs and we did identify examples where more than one motif is bound for both homo and heterodimers. We report the MEME motifs in [Supplementary-material SD4-data] (see below).

*Certain bZIP subfamilies have been known for many years to bind more than 1 type of motif. For example, see early work from Kevin Struhl's lab on Ap-1 versus ATF/CREB bZIP domains: Ap-1 proteins recognize overlapping TGAC half-sites TGA(C|G)TCA, while ATF/CREB proteins recognize adjacent half-sites TGACGTCA; however, "AP-1 proteins prefer to bind AP-1 sites, but they also bind ATF/CREB sites with only slightly lower affinity" (PMID: 7630732). Given that the authors use single motifs to represent the specificity of homo and heterodimers, such preferences for more than one arrangement of half-sites is not illustrated in the paper.*

We now cite the groundbreaking study from the Struhl lab in our revised manuscript to provide better context for our work.

In agreement with the earlier studies, we do identify homo and heterodimers that bind multiple motifs, and such cases are marked in [Supplementary-material SD4-data]. In Figure 2, we present a specificity and energy landscape (SEL) of JUN•ATF3 and highlight the ability of one dimer to bind multiple different binding sites. Examination of SELs for all 102 bZIPs reveals the extent to which different bZIPs associate with multiple motifs. In our revised SEL pipeline, we now include the feature to identify motifs within the sequences that contribute to peaks in a SEL plot. Given the non-uniform impact of changes in the core half-site or in the flanking sequences on the affinity of different bZIP dimers, simply listing all the arrangements results in tables that have the potential to obscure meaningful differences.

In response to the reviewers’ suggestion that we illustrate preferences of bZIPs for different half-site arrangements, we now include Figure 2, wherein we display the relative affinity of all 102 dimers for half-sites that occur in one of the six classic motifs (second set of six rows). We also display the relative affinity of all 102 bZIPs for sequences identified from SELs of JUN•ATF3 and ATF4•CEBPG (first set of 10 rows in Figure 2). Together, this heatmap illustrates the relative affinities of 102 bZIP dimers for the entire CRE (row 1), TRE (row 2), CRE-CAAT (rows 6 & 7), TRE-CAAT (row 10) sites as well as the affinities for half-sites from homodimers binding to the six classic sites (rows 11-16). We respectfully note that expanding this heatmap to include all possible half-site arrangements results in a complex table that is less accessible than SELs and that misses many subtle, yet meaningful differences between dimers (such as the differential impact of flanking sequences on the binding of CREB3 and CREB1 to the CRE site -row 5 or the non-uniform impact of changes in the core CAAT half-site that impacts different CEBP containing heterodimers to different degrees –row 8). We propose that SELs plots together with affinity-based motif lists would most effectively capture the range of sequences that can be targeted by a given transcription factor.

*2) From the manuscript it seems clear that if we take only the top motif for each dimer, then "emergent" motifs cannot be predicted from the motifs of the individual homodimers. But if for each homodimer we consider all the motifs they recognize (sometimes with different affinities, other times with very similar affinity), can we then predict some of the "emergent" motifs?*

*On a related note: The authors mention that "Nine out of the 80 heterodimers (11%) enriched two motifs ([Supplementary-material SD4-data]). For example, BATF•CEBPG enriched both CRE-CAAT and CRE-L-CAAT motifs." But what about homodimers? Some homodimers might also bind two motifs.*

We use three approaches to identify and validate binding motifs for both homo and heterodimers. First, when applying motif-finding algorithms we do not limit the search to a single motif. Second, rather than focusing on top binding sites, we examine the entire bZIP-DNA interactome with SELs. As mentioned above, in SELs alternate motifs emerge as peaks in the mismatch rings (see Figure 2). Finally, we use EMSA-FRET to compare and independently validate the sequence preferences of homo- and heterodimers (Figure 3).

In Figure 3, we found BATF3 homodimer binds CRE sites but displays no binding to the CRE-L site. By comparison, the heterodimers formed by BATF3 (such as ATF2•BATF3 and BATF3•JUN), bound both CRE and CRE-L sites comparably. XBP provides an excellent example of a homodimer binding to both CRE and CRE-L sites with comparable affinity (Figure 3) and yields both motifs ([Supplementary-material SD4-data]). Moreover, JUN homodimer binds to CRE while displaying no detectable affinity for CRE-L (Figure 3).

We also note that while homodimers may weakly associate with some sequences, the heterodimerization-dependent switch in their ability to bind the same sequence with high affinity cannot predicted by current models of protein-DNA recognition. Moreover, it is unclear why one weak site, over many others, would emerge as the heterodimer-preferred site. Therefore, while a list of possible sites may be compiled by mixing and matching half-sites, it is not possible to a priori identify emergent sites as the most preferred sites of heterodimers. For this reason, we term them emergent sites as they arise due to heterodimer formation.

*The authors tried to address this issue by using "specificity and energy landscapes" (SEL). While I understand that these landscapes contain a lot more information than the motifs, they are not intuitive to interpret and it is not clear to me how one would immediately extract information about multiple half-site arrangements from SELs. A much simpler representation would be by using several motifs for each homo/hetero-dimer that binds more than 1 half-site arrangement.*

While the use of SELs to examine protein-DNA interactomes is not common, SELs do provide a comprehensive representation of the sequence specificity a DNA binding protein. In fact, many of the emergent sites are not captured by motif finding algorithms and are often compressed into more generic/classic motifs. Our SEL-based evaluation of various publicly available protein-DNA interactomes revealed up to 40% had compressed multiple related motifs into low-information content generic motifs (Carlson et al. *PNAS* 2010). Furthermore, the contribution of flanking sequences to binding even the most ideal consensus motifs is greatly dampened by most motif finding algorithms (as noted by us in publications from 2006 onwards and by others –Gordon et al. *Mol Cell* 2013). Thus, we request the reviewers consider our inclusion of the heatmap (Figure 2) alongside our SEL displays as scholarly effort to increase the general awareness and appreciation of the breadth of cognate sequences bound by DNA (and RNA) binding proteins.

*3) Comparing SELs is also difficult. When two dimers have a site in common, how can we see that from the SELs? Unless the seed is exactly the same, the SELs are hard to compare.*

Indeed, as the reviewer points out, the best way to compare two dimers is to make landscapes for each bZIP using the same seed; this is easy to do, and we have found it very effective. Landscapes generated using the same seed permit the generation of a difference-SEL that readily reveals differential specificities of closely related or completely unrelated DNA binding proteins (for example, see Erwin et al., PNAS 2016). This reviewer’s point has prompted us to formalize the “differential-SEL” analysis and disseminate it via a publication in a bioinformatics methods-focused journal.

*The authors interpret some SELs in subsection “Specificity and Energy Landscapes (SELs) reveal the entire spectrum of cognate sites bound by heterodimers”, but couldn't the same insights be gained by simply deriving more than 1 motif from each data set?*

*Maybe one possibility for presenting the data is to generate a set of motifs representing all the half-site combinations observed in the data, and then for each dimer plot its affinities for sites matching each motif.*

As we mention above, our examination of publicly available protein-DNA interactomes, we find that infrequent binding sites as well as many closely related binding sites are often compressed into binding motifs that mask information content. That being said, we recognize that PWMs represent a simpler way to represent particular motifs, and one that is familiar to the wider community. Thus, we propose the inclusion of motifs alongside SELs, as we present in [Supplementary-material SD4-data], as a good strategy for representing the data. Note that the entries for several bZIPs in [Supplementary-material SD4-data] do include two PWM motifs (grouped in brackets).

*4) Few studies of heterodimers binding to DNA carefully dissect the binding of hetero- versus homodimers. The authors did this using EMSA-FRET assays, and for a large number of heterodimers (67).*

*Although this data is very valuable, the way it is presented is not clear – see Figure 3 and associated text. I would recommend using a single color for PPIs, and a single color for PDIs (with intensity proportional to affinity of the interactions). Or even have separate plots for PPIs versus PDIs. Alternatively, the authors could show the data as barplots with PDI strength above the x-axis (i.e. as positive numbers) and PPIs underneath the x-axis (i.e. as negative numbers). Another option would be to use circles where the size reflects PDI strength and the color intensity reflects PDI strength. In subsection “EMSA-FRET analysis to validate heterodimer binding to different cognate sites” the authors say: "It is readily evident that neither the JUN nor the ATF3 homodimers associate with the emergent site identified by SELs for JUN•ATF3 (Figure 3 and Figure 3—figure supplement 1)." To this reviewer it was not evident, and it took quite a lot of time to extract this information from SELs and the figures in the manuscript.*

Based on the reviewer’s advice, we have completely revised Figure 3 and now display the data as bar graph using the color and size scales suggested by the reviewer.

*5) The case study of ATF3's specificity being influenced by partner TFs is clear and interesting.*

We thank the reviewer for their appreciation of this study.

*6) Regarding the* in vivo *ATF3 analysis: the authors found that the motifs of different ATF3 dimes are best at explaining the ATF3 ChIP-seq data in different cell lines. Did the authors verify that the partners identified by their analysis are actually expressed, at high enough levels, in those cell lines?*

To address this concern, we examined RNA-seq data from the ENCODE project to verify that all bZIP genes described in Figure 5 are expressed in the ENCODE cell lines used in our analysis (i.e., H1, HEPG2, and K526). With the exception of BATF and BATF3 the remaining bZIP monomers are expressed in these cell lines. In addition, RNA-seq data from the van Lohuizen laboratory (Gargiulo et al., 2013) shows that the bZIP genes used for the analysis in Figure 5 are expressed in the GBM1 cell line that was used for ChIP-seq of ATF3. We now include this information in section Heterodimer sites enriched in SELEX-seq map to occupied genomic loci in vivo: “We used published RNA-seq datasets to verify the expression of the bZIP genes used for the ATF3 heterodimer analysis ([Supplementary-material SD3-data]).”

*Also, are the reported differences in AUC-ROC values between different motifs (Figure 5) significant? What is the magnitude of differences observed for motifs trained on replicate SELEX-seq experiments? The AUC-ROC values are generally not robust to small changes in the motifs. The analysis shown in Figure 5, based on the SELEX-seq data, better reflects the cell-type specific differences in ATF3 heterodimer binding.*

We fully concur with the reviewer that AUC-ROC values are not robust to small changes. We included those analyses to conform to the common practices in the field. Moreover, while we display the original AUC-ROC curves in Figure 5—figure supplement 1 and report AUC values in [Supplementary-material SD3-data], we have removed the table from Figure 5.

7) The analysis in the last section of Results (Figure 7) is very interesting. However, the "noise factor" used in the analyses of SNP data seems ad-hoc. What is the rationale for the formula used to calculate the noise factor?

In the SNP analysis, we calculated a predicted fold-change between the CSI score for a reference allele (from hg19) and the CSI score for the corresponding alternate allele using the formula described in the Single Nucleotide Polymorphism (SNP) scoring in Materials and methods. To make our fold-difference predictions robust, we added a “noise factor” to the CSI intensity in the numerator and the denominator. The rationale of the noise factor is to make the prediction less sensitive to calculating ratios with low CSI scores. Instead of adding a constant value to every CSI intensity to mitigate for low CSI intensities, we added a value (*A*) that varied according to the range of the CSI intensities for a given dimer. This was done to take into account the differences in the observed range of CSI intensities between dimers.